# Predictions enable top-down pattern separation in the macaque face-processing hierarchy

Tarana Nigam [1,2,3,4] & Caspar M. Schwiedrzik [1,2,3] ✉

Distinguishing faces requires well distinguishable neural activity patterns. Contextual information may separate neural representations, leading to enhanced identity recognition. Here, we use functional magnetic resonance imaging to investigate how predictions derived from contextual information affect the separability of neural activity patterns in the macaque face-processing system, a 3-level processing hierarchy in ventral visual cortex. We find that in the presence of predictions, early stages of this hierarchy exhibit well separable and high-dimensional neural geometries resembling those at the top of the hierarchy. This is accompanied by a systematic shift of tuning properties from higher to lower areas, endowing lower areas with higher-order, invariant representations instead of their feedforward tuning properties. Thus, top-down signals dynamically transform neural representations of faces into separable and high-dimensional neural geometries. Our results provide evidence how predictive context transforms flexible representational spaces to optimally use the computational resources provided by cortical processing hierarchies for better and faster distinction of facial identities.

We have all mistaken someone or been mistaken for somebody else. Indeed, recognizing people by their face is a significant computational challenge both for humans and machines, especially in light of variations in appearance, e.g., with a different hairstyle or pose. This is particularly evident when distinguishing lookalikes (or doppelgänger)[1] or identical twins[2] (Fig. 1A). To do so, we need additional *contextual* information, e.g., associations from previous encounters, which we use to make predictions[3]. Predictions render recognition better and faster, yet the computations and neural implementation mediating those effects remain poorly understood. In particular, it is unknown how contextual predictions optimize sensory neural representations, especially for distinguishing individuals by their face.

The reason for why we struggle to distinguish doppelgänger or twins is that similar stimuli lead to correlated neural activity patterns. To distinguish stimuli like faces, neural representations should have minimal overlap and hence be "separable". Separability is largely determined by stimulus properties, i.e., how physically similar faces are, and only achieved at hierarchically higher areas in the ventral visual stream, where neural representations are also robust against variations like pose change[4]. How can incorporating predictive information enable distinguishing identities? Predictive processing theories suggest that higher-order areas generate predictions that are communicated to lower areas via feedback pathways[5]. We hypothesized that predictions reflect the representations of the areas that generate the predictions, i.e., of higher-level areas. When they are transmitted from higher-order to lower areas, they carry this computational format - hence, incorporating higher-order predictive information should lead to more separable

[1]Neural Circuits and Cognition Lab, European Neuroscience Institute Göttingen – A Joint Initiative of the University Medical Center Göttingen and the Max Planck Institute for Multidisciplinary Sciences, Grisebachstraße 5, 37077 Göttingen, Germany. [2]Perception and Plasticity Group, German Primate Center – Leibniz Institute for Primate Research, Kellnerweg 4, 37077 Göttingen, Germany. [3]Leibniz ScienceCampus 'Primate Cognition', Göttingen, Germany. [4]International Max Planck Research School 'Neurosciences', Georg August University Göttingen, Grisebachstraße 5, 37077 Göttingen, Germany. ✉e-mail: c.schwiedrzik@eni-g.de

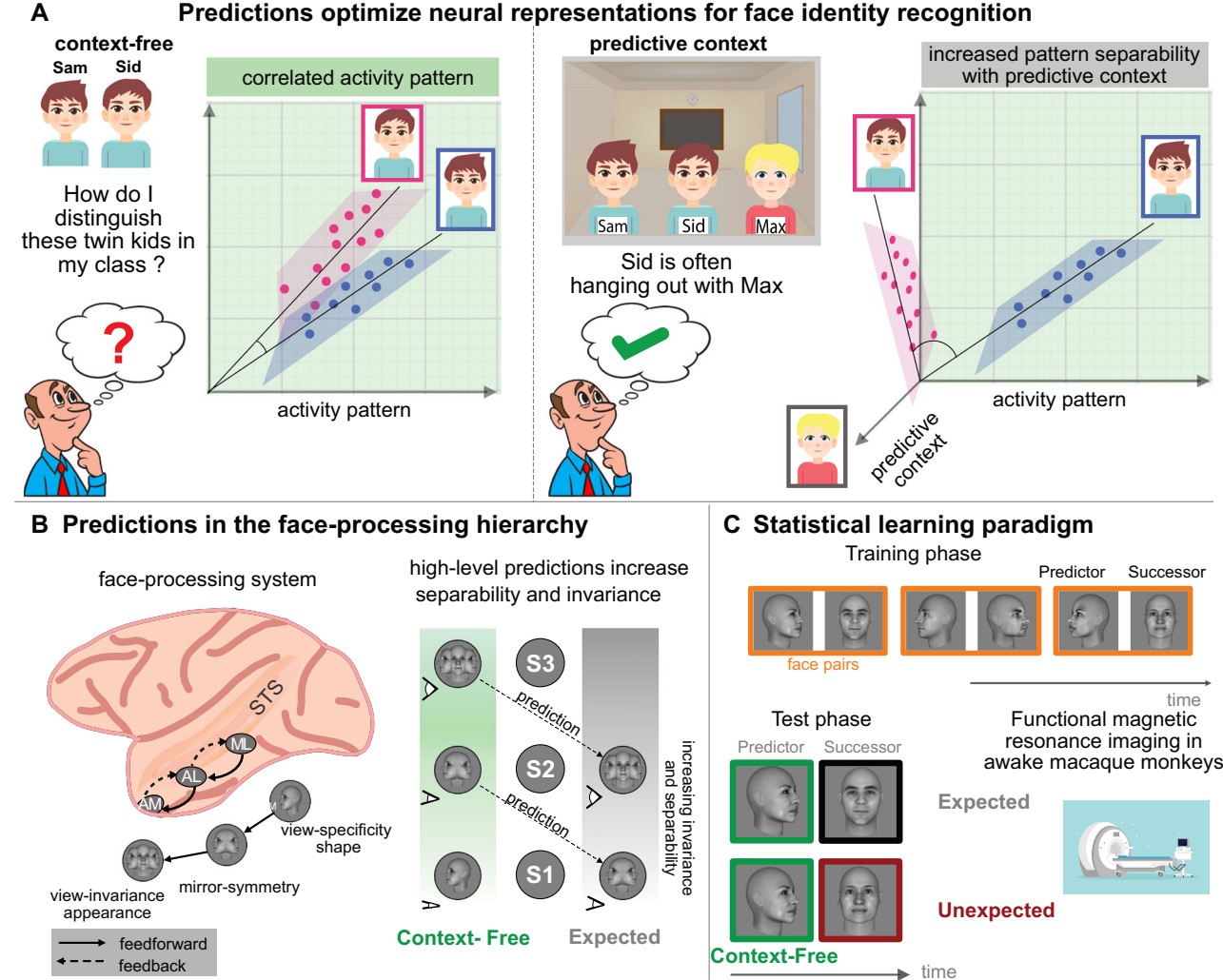

**Fig. 1 | Increasing pattern separability through top-down predictions. A** Similar faces, like those of lookalikes and twins are hard to distinguish because they lead to poorly separable neural activity patterns. Contextual predictions, derived, e.g., from previously learnt associations, can aid distinguishing similar stimuli. We hypothesize that predictive context facilitates recognition by increasing separability between neural activity patterns. E.g., a teacher who has trouble distinguishing identical twins in his class may utilize learnt contextual associations like friends of one of the twins to help distinguish the similarly looking faces. **B** In the face processing system, the lowest area ML has view-specific shape representations, the intermediate area AL has mirror-symmetric representations, and the top of the hierarchy, AM, has view-invariant appearance representations. All stages are directly and reciprocally connected. Higher face areas contain neural representations with minimal overlap (high separability) and view-invariance. We hypothesize that incorporating high-level predictive information (via feedback pathways) increases invariance and separability in lower areas, **C** To create contextual predictions, we exposed monkeys to face pairs in a statistical learning paradigm. Once learned, the first face in a pair (the "predictor") predicts the second face in a pair (the "successor"). In the test phase, we presented predictable face pairs (60% of all trials) along with their violations during fMRI. Artwork depicting faces of kids in panel A by M. Shitik and is reproduced with permission by the artist. The image of the teacher is reproduced with permission from Alamy. The face images in panels B and C were created using FaceGen Modeller.

(Fig. 1A) and invariant (Fig. 1B right) representations already in lower areas of the hierarchy.

To investigate how predictability optimizes neural representations for distinguishing identities, we turned to the macaque face-processing system (Fig. 1B left). This information processing hierarchy in inferotemporal (IT) cortex is highly specialized for faces, and the neural representations and transformations across successive stages of processing are well understood. In the ventral part of this system, representations become successively invariant to face view while the system extracts facial identity: the lowest area, ML, represents face shape and possesses view-specificity; the intermediate area AL has mirror-symmetric representations; at the apex of the hierarchy, in AM, representations are view-invariant and contain facial appearance information for identity recognition[6,7]. We used functional magnetic resonance imaging (fMRI) to access these qualitatively distinct representations in all three face-areas simultaneously and to compare them to the established electrophysiological ground-truth. By training animals to expect specific faces based on temporally preceding information[8] (Fig. 1C), we could test how predictions affect neural representations, and whether feedback pathways are involved.

We find a gradient of pattern-separability along the face-processing hierarchy with the highest separability at the top, in AM, even when no predictive context exists. Once predictions come into play, pattern-separability increases in lower stages of the hierarchy. This is accompanied by the emergence of abstract, view-invariant representations at these lower processing stages: rather than the view-specific tuning properties they show in the absence of predictions, we find that lower face-areas express representations characteristic of higher stages of processing when stimuli are predictable. This suggests that high-level prediction signals cascading-down the face-processing system dynamically transform neural representational spaces, endowing them with increased pattern-separability and invariance.

This may enable lower areas to abstract from irrelevant features for enhanced distinction of identities. Our findings highlight the flexibility of neural representations in IT and suggest a specific mechanism, top-down pattern-separability, by which predictions exert their effects.

## Results

As a first step in investigating whether and how predictions alter neural coding in the face-processing system, we localized three face-areas - ML, AL, and AM – in two macaque monkeys using a standard face localizer for whole-brain fMRI (Supplementary Fig. S1). We then trained the monkeys to associate temporally contiguous faces into pairs using statistical learning[8] (Fig. 1C). Once learned, the first face in a pair (the "predictor") fully predicts the second face (the "successor"). We exposed the monkeys to nine face-pairs for a minimum of three weeks, providing ample time to elicit effects of predictability in IT cortex[9]. To assess the acquisition of face-pair associations, we evaluated pupil dynamics, an established marker of statistical learning[10]. Indeed, we found pupil entrainment to the frequency of the face-pairs after training (Supplementary Fig. S2). Once we established face-pair learning in the monkeys, we investigated if and how coding spaces changed as a function of contextual predictions across the face-processing system using a rapid event-related design for fMRI (Fig. 1C). In the *expected* condition (60% of trials), we presented learned face-pairs, such that predictions were elicited once the predictor had been shown and thus took effect when the successor appeared. On violation trials, predictions went unfulfilled as we presented an *unexpected* identity in the expected view or an unexpected view of the expected identity as the successor[8]. Since our focus was on investigating how predictability contributes to the identity recognition, one of the main computational goals of face-processing, we mostly concentrate on conditions involving facial identity.

### Predictions increase separability in the face-processing hierarchy

Before evaluating how predictive context optimizes neural representations for face individuation, we first established how neural separability for identity increases across the face-processing hierarchy in the absence of predictive context (the *context-free* condition) using Representational Similarity Analysis (RSA)[11]. In each of the three face-areas we measured cosine distances between activity patterns elicited by different faces (Fig. 2Ai). We report average results because these distances were highly correlated across monkeys (Spearman's rho(7) = 0.95, $p = 3.5e{-}04$). Pattern-separability increased along the hierarchy (ML < AL < AM, paired Hotelling test, all $F_{(2,34)} > 56.7582$, all $p < 1.462e{-}11$, corrected for multiple-comparisons). This corroborates previous electrophysiological studies in IT cortex outside the face-processing system[12] and indicates that, in the absence of predictability, high levels of pattern-separability for facial identities are only achieved in AM, the top of the face-processing hierarchy.

After determining baseline pattern-separability across the face-processing system, we evaluated whether predictability aids face identification by increasing pattern-separability in lower areas. We compared separability across face identities between the expected and context-free conditions (Fig. 2Aii): when faces were expected, we found that lower areas in the face-processing hierarchy, ML and AL, increased pattern-separability for identities by ~10° from 52.1° to 61° in ML and 67.7° to 77.2° in AL (Fig. 2B; William-Watson test, ML: $F_{(1,70)} = 79.74$, $p = 1.067e{-}12$; AL: $F_{(1,70)} = 34.49$; $p = 2.619e{-}07$, corrected for multiple comparisons). In contrast, pattern-separability in the higher area AM remained unchanged (84.7° vs. 87.5°; $F_{(1,70)} = 3.53$; $p = 0.0644$, corrected for multiple comparisons). This suggests that high separability otherwise characteristic of the top of the processing hierarchy can be approached in lower areas if predictive information exists - potentially facilitating readout of information from well decorrelated representations.

Predictive processing theories suggest that predictions serve to compute prediction errors (PE) as a deviation between the prediction and the sensory input to reduce redundancy in information transmission[13]: redundant information is removed, optimizing the dynamic range of neurons and allowing them to efficiently signal decorrelated or unpredicted information[14]. In terms of pattern separation, this may entail higher pattern-separability when PEs (to unexpected faces) occur, possibly even higher than in the expected condition where no PEs need to be computed. To evaluate this possibility, we compared pattern separation between the violation and context-free conditions in the three face areas. Prediction violations increased pattern-separability in the entire face-processing hierarchy (Fig. 2B; William-Watson test, ML: 77.7° vs. 52.1°, $F_{(1,70)} = 485.84$, $p < 0.0001$; AL: 82.8° vs. 67.7°, $F_{(1,70)} = 115.65$, $p = 4.44e{-}16$; AM: 89.6° vs. 84.7°, $F_{(1,70)} = 14.00$, $p = 0.0003$, corrected for multiple comparisons). AM patterns were, in fact, almost orthogonal in this condition (mean 89.6°, one-sample test for mean angle vs. 90°, $p > 0.05$). Interestingly, prediction violations increased pattern-separability even beyond the separability achieved in predictable conditions, but only in lower stages of the hierarchy (paired Hotelling test, ML: 77.1° vs. 61°, $F_{(2,34)} = 97.0819$, $p = 2.631e{-}14$; AL: 82.8° vs. 77.2°, $F_{(2,34)} = 9.9190$, $p = 8.084e{-}04$; AM: 89.6° vs. 87.5°, $F_{(2,34)} = 1.7520$, $p = 0.1887$, corrected for multiple-comparisons). PEs thus appear to enhance pattern-separability in lower processing stages, possibly by optimizing the available dynamic range, while the top maintains a stable prior. Taken together, our results suggest that pattern separation may be the mechanism underlying the beneficial effects of predictability in IT cortex, decreasing interference between representations of identities. By increasing pattern-separability at lower stages of processing, predictions may thus help in achieving one of the core computations for face/object recognition earlier than in the absence of predictive context.

### High-dimensional neural codes underlie predictability-induced increase in separability

How do predictions give rise to increased separability in the hierarchy? We hypothesized that contextual information provided by predictions could add feature dimensions for separability – implying that predictive cues improve separability by increasing dimensionality of neural representational space (Fig. 2Aiii). To evaluate this hypothesis, we quantified dimensionality using the Participation Ratio (PR)[15,16] which describes how evenly the variance of neural activity is spread (i.e., dimensionality is high if variance is spread across all dimensions). PR was computed per face area and condition across all voxels and stimuli. We first established that PR was interpretable, i.e., well differentiated from noise, in all conditions and face-areas (Fig. 2C; comparison to noise-ceiling for each monkey, all $p < 9.99e{-}04$, 1000 permutations, corrected for multiple comparisons). Next, we investigated PR as a function of hierarchy. In the context-free condition, we found higher dimensionality in AM than in lower stages ML and AL (comparison of difference to noise ceiling between areas for each monkey, all $p < 9.9e{-}04$, 1000 permutations, corrected for multiple comparisons). For expected faces, dimensionality increased in lower areas AL and ML (that had shown increases in pattern separability) compared to the context-free condition (Monkey L – ML: $p = 0.0189$, AL: $p = 0.004$; Monkey P – ML: $p = 0.045$, AL: $p < 0.0001$, 1000 permutations, corrected for multiple comparisons). When expectations were violated, dimensionality increased even further compared to the expected condition in ML (Monkey L: $p < 0.0001$; Monkey P: $p < 0.0001$, 1000 permutations, corrected for multiple comparisons) and AL (Monkey L: $p < 0.0001$; Monkey P: $p = 0.0001$, 1000 permutations, corrected for multiple comparisons), and slightly less consistently in AM (Monkey P: $p < 0.0001$; Monkey L: $p = 0.055$, 1000 permutations, corrected for multiple comparisons). Dimensionality and separability across conditions and hierarchy were highly

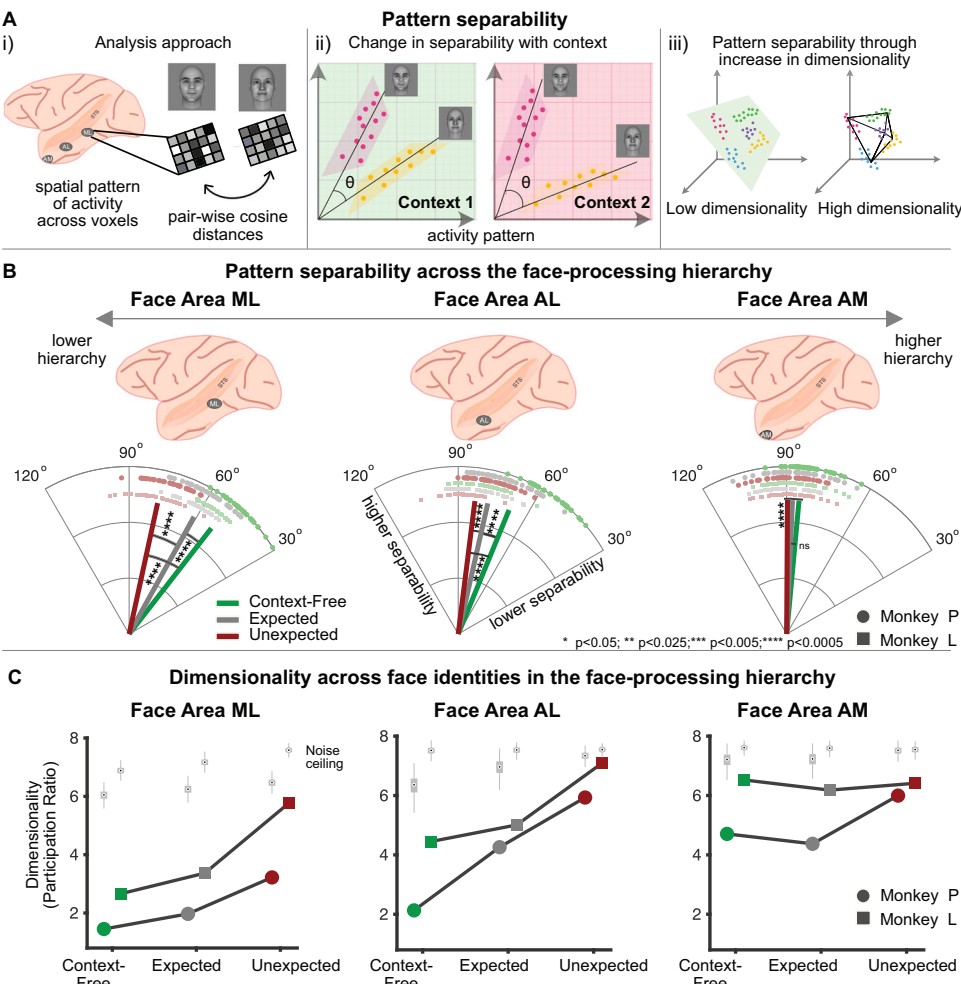

**Fig. 2 | Predictions increase separability in the face-processing hierarchy. A.i** Pattern-separability was computed as pair-wise cosine distances between activity patterns elicited by all faces presented in each context. **A.ii** Neural activity patterns are well separable when there is minimal overlap between population vectors. Separability between the same faces can depend on context. The angle θ between population vectors denotes separability. **A.iii** Activity patterns may not be well-separable in low-dimensional spaces. Expanding dimensionality, e.g., by adding feature dimensions, can increase separability. **B** Pattern-separability increases along the face-processing hierarchy from ML (52.1°) to AL (67.1°) to AM (84.7°) in the context-free condition (green). Lower areas in the hierarchy increase pattern-separability when faces are expected (gray; ML 52.1° to 61°; F(1,70) = 79.74, $p = 1.067e\text{-}12$; AL: 67.7° to 77.2°, F(1,70) = 34.49, $p = 2.619e\text{-}07$, William-Watson test), whereas AM shows no difference in separability (84.7° vs. 87.5°, F(1,70) = 3.53, $p = 0.0644$). Prediction violations (red) increase separability in all areas even further (ML: 77.7° vs. 52.1°, F(1,70) = 485.84, $p < 0.0001$; AL: 82.8° vs. 67.7°, F(1,70) = 115.65, $p = 4.44e\text{-}16$; AM: 89.6° vs. 84.7°, F(1,70) = 14.00, $p = 0.0003$, William-Watson test). Only lower areas have higher separability in the unexpected than in the expected condition (ML: 77.1° vs. 61°, F(2,34) = 97.0819, $p = 2.631e\text{-}14$; AL: 82.8° vs. 77.2°, F(2,34) = 9.9190, $p = 8.084e\text{-}04$; AM: 89.6° vs. 87.5°, F(2,34) = 1.7520, $p = 0.1887$, paired Hotelling test). Lines depict mean angles,

squares and circles pair-wise angles between identities per monkey. $n = 36$ pair-wise angles. All $p$-values were corrected for multiple comparisons and all tests are two-sided. **C** Estimated dimensionality in all areas and conditions is lower than that of the noise ceiling (light gray boxplots, all $p < 9.99e\text{-}04$, permutation test (one-sided), 1000 permutations). In the context-free condition (green), dimensionality is higher in AM than in lower areas (all $p < 9.9e\text{-}04$ for both monkeys, permutation test (two-sided), 1000 permutations). Dimensionality increases when faces are expected (gray) in ML (Monkey L: $p = 0.0189$, Monkey P: $p = 0.045$) and AL (Monkey L: $p = 0.004$; Monkey P: $p < 0.0001$). Violations (red) expand dimensionality further in ML (Monkey L: $p < 0.0001$, Monkey P: $p < 0.0001$) and AL (Monkey L: $p < 0.0001$; Monkey P: $p = 0.0001$), and less consistently in AM (Monkey P: $p < 0.0001$; Monkey L: $p = 0.055$). Squares and circles depict individual monkeys. All $p$-values were corrected for multiple comparisons. The permutation tests to test whether the estimated dimensionality is lesser than noise ceiling are one-sided, all other tests to compare conditions or ROIs are two-sided. See Supplementary Table 1 for each value of estimated dimensionality (participation ratio) and the distribution of the noise ceiling depicted in the light gray boxplots (minima, maxima, median, bound box: 25th, 75th percentile and whiskers: lower and upper adjacent). Source data are provided as a Source data file. The face images in panel A were created using FaceGen Modeller.

correlated (Spearman's rho(7) = 0.95, $p = 3.5273e\text{-}04$), suggesting that contextual signals (like predictions and/or PEs) increase pattern-separability through an expansion of representational dimensionality.

Previous studies have shown that deviance between predictions and sensory inputs is signaled as a difference in the magnitude of neural activity between unexpected and expected stimuli[8,17,18]. Indeed, we also found stronger responses in the unexpected than in the expected condition in all three face areas (Supplementary Fig. S3; all $p < 3.5e\text{-}04$). However, this difference in magnitude did not explain the

increase in pattern-separability (correlation between mean separability and mean amplitude across face areas and conditions: Spearman's rho(7) = 0.0167, $p = 0.9816$; correlation between stimulus-wise separability and magnitude for each condition and face area separately: highest Spearman's rho(7) = 0.10, all $p > 0.05$) nor dimensionality (rho(7) = 0.2667, $p = 0.4933$) with contextual information across the hierarchy. This is in line with previous findings indicating that population magnitude and separability in IT cortex convey different information[19].

## High-level predictions cascade down the face-processing hierarchy

How can predictive signals increase separability in lower stages of the hierarchy? One possibility is that lower areas inherit (more separable) representations from higher areas via feedback pathways conveying predictions. Higher areas in the ventral stream contain representations that are not only highly separable, but also invariant, e.g., for view. A consequence of predictions being passed from higher to lower areas is that lower areas should inherit the invariant tuning properties of the areas where the predictions originate. To evaluate this hypothesis, we determined the representational geometry across the face-processing hierarchy using RSA[11] (Fig. 3Ai). We computed cosine distances between the activity patterns for different face stimuli and compared 1st level RDMs to model RDMs based on known electrophysiological tuning properties of each face area (Fig. 3Aiii, see below). Results were highly correlated across monkeys (Spearman's rho(7) = 0.5524; $p = 2.3830e{-}04$), hence, we report results from averaged 1st level RDMs.

First, in the context-free condition, we established that our approach recapitulated the known electrophysiological tuning properties of face-processing areas[6,7]. As expected, the known tuning properties best explained the representational geometry in all three face areas: view-specific representation of facial shape in ML, mirror-symmetric representation in AL, and view-invariant representation of facial appearance in AM (Fig. 3B; bootstrapped test comparing 2nd level correlation coefficients, one-sided, 10000 bootstrap samples; ML: shape vs. 3 others, difference = 0.3153; $p = 0.0417$; AL: mirror symmetry vs. 3 others, difference = 0.8364, $p < 0.0001$; AM: 2 appearance-based models vs. 2 others, difference = 0.4712, $p = 0.0019$).

Next, we tested the hypothesis that under predictable conditions, lower areas exhibit higher-order tuning properties. To this end, we compared representational geometry between the expected and context-free conditions in ML and AL. These lower areas indeed expressed higher-order, abstract representations: ML showed an increase in mirror-symmetric tuning, characteristic of AL (one-sided Raghunathan's test, $z = -1.684$, $p = 0.0460$), while AL showed increased tuning for appearance ($z = -2.446$, $p = 0.0072$) and view-independent appearance ($z = -4.290$, $p = 8.941e{-}6$), characteristics of AM. Moreover, in the predictable condition we observed a decrease in the classical, feedforward tuning properties in the entire face-processing network, i.e., decreased shape tuning in ML ($z = 1.848$, $p = 0.0323$), decreased mirror-symmetric tuning in AL ($z = 2.441$, $p = 0.0073$), and decreased appearance ($z = 2.363$, $p = 0.0091$) and view-independent appearance tuning ($z = 2.626$, $p = 0.0043$) in AM. Thus, rather than expressing their own feedforward tuning properties, lower areas ML and AL exhibited representations characteristic of higher face-areas when stimuli were predictable.

Furthermore, we found higher functional pattern connectivity between all pairs of face areas in the expected compared to context-free condition (difference between correlations = 0.2143, $p = 0.048$), suggesting that more information is exchanged between face areas in the presence of predictions. Taken together, this suggests that high-level predictions cascade-down the entire face-processing hierarchy along feedback pathways, carrying with them a representational format that enables better pattern separation.

Finally, we investigated the representational format of prediction errors (PEs) in the unexpected conditions. We operate under the premise that PEs in high-level cortex do not merely signaling surprise, but are teaching signals[20] carrying feature-specific content[8,21,22]. Our results so far suggest that lower areas inherit tuning properties from higher areas through predictions. Because according to predictive processing theory, PEs are computed relative to predictions generated in higher areas, we hypothesized that PEs in lower areas should also express higher-level tuning properties[8,23–25]. Alternatively, PEs could enhance feedforward information, i.e., the local tuning properties of the area in which they are generated[26]. To differentiate these hypotheses, we isolated PEs by computing the relative difference between the expected and the unexpected conditions[27] and then applied RSA (Fig. 3Aii). For identity PEs, we found that the pattern of tuning properties was highly correlated with the expected condition (Fig. 3B; Spearman's rho(7) = 0.7273, $p = 0.01$), suggesting that the representational geometry evident in the expected condition indeed forms the basis of the computation of PEs when unexpected identities occur. Further mirroring the expected condition, there was an increase in appearance (one-sided Raghunathan's test, $z = -2.383$, $p = 0.0086$) and view-independent appearance tuning ($z = -2.412$, $p = 0.0079$) in AL compared to the context-free condition, concomitant to a decrease in mirror-symmetric tuning ($z = 3.516$, $p = 0.0002$). Similarly, appearance ($z = 2.102$, $p = 0.0178$) and view-independent appearance tuning ($z = 2.577$, $p = 0.0049$) again decreased in AM. This suggests that identity PEs in the unexpected identity condition are computed in AL on the basis of appearance tuning originating in AM, i.e., one stage earlier than identity information in the context-free condition. Similarly, and in accordance with our previous electrophysiological results from ML[8], we found significant mirror-symmetric tuning in ML (bootstrapped test, $p = 0.039$, n = 10000 bootstrap samples) in the unexpected view condition (although this form of tuning did not exhibit a statistically significant increase over the context-free condition; one-sided Fisher test, $z = -0.887$, $p = 0.188$). This was accompanied by a decrease of mirror-symmetric tuning in AL ($z = 3.479$, $p = 0.0003$). Overall, this suggests that PEs are computed on the basis of high-level prediction signals in lower areas of the face-processing hierarchy.

## Discussion

Taken together, our results reveal that incorporating predictive cues optimizes neural representations for distinguishing facial identities by increasing separability of neural activity patterns already early in the face-processing hierarchy. The improved separability with contextual information comes with an expansion of dimensionality of neural representational space. These effects can be achieved by cascading-down prediction signals via feedback pathways - such that well separated, invariant representations characteristic of higher face-areas are passed down to lower areas. Our work thus proposes a distinct mechanism, i.e., top-down pattern-separability, through which predictions exert their effects.

Well separated neural representations are beneficial for processing because they allow for linear readout of information[28,29] and learning new associations[30]. Pattern separation is a well-established computational mechanism, originally attributed to hippocampal and cerebellar circuits[31–33]. Recent proposals assign this computation also to cortex[34] - albeit with different implementations owing to the distinct architectures. We find that predictive information enables high separability in lower areas of the ventral visual stream, which is otherwise only characteristic of the top of this cortical processing hierarchy. Highly separated representations early on in the hierarchy may be beneficial since they facilitate early readout of high-level information like object identity.

Theory suggests that highly separable representations can arise from dimensionality expansion[28,29] - providing a larger neural activity space for linear separability. We find that AM at the top of the face-processing hierarchy, which is thought to directly contribute to face individuation, has high dimensionality, similar to the human face-processing hierarchy[35]. This may be related to the inherently high dimensionality of face space required to differentiate individuals[36]. Increases in separability in AL and ML brought about by predictability are accompanied by an expansion of dimensionality. This suggests that the dimensionality of IT cortex activity is in principle flexible and not hardwired. Our findings thus resonate with studies in other species and brain areas which have shown that the dimensionality of neural activity

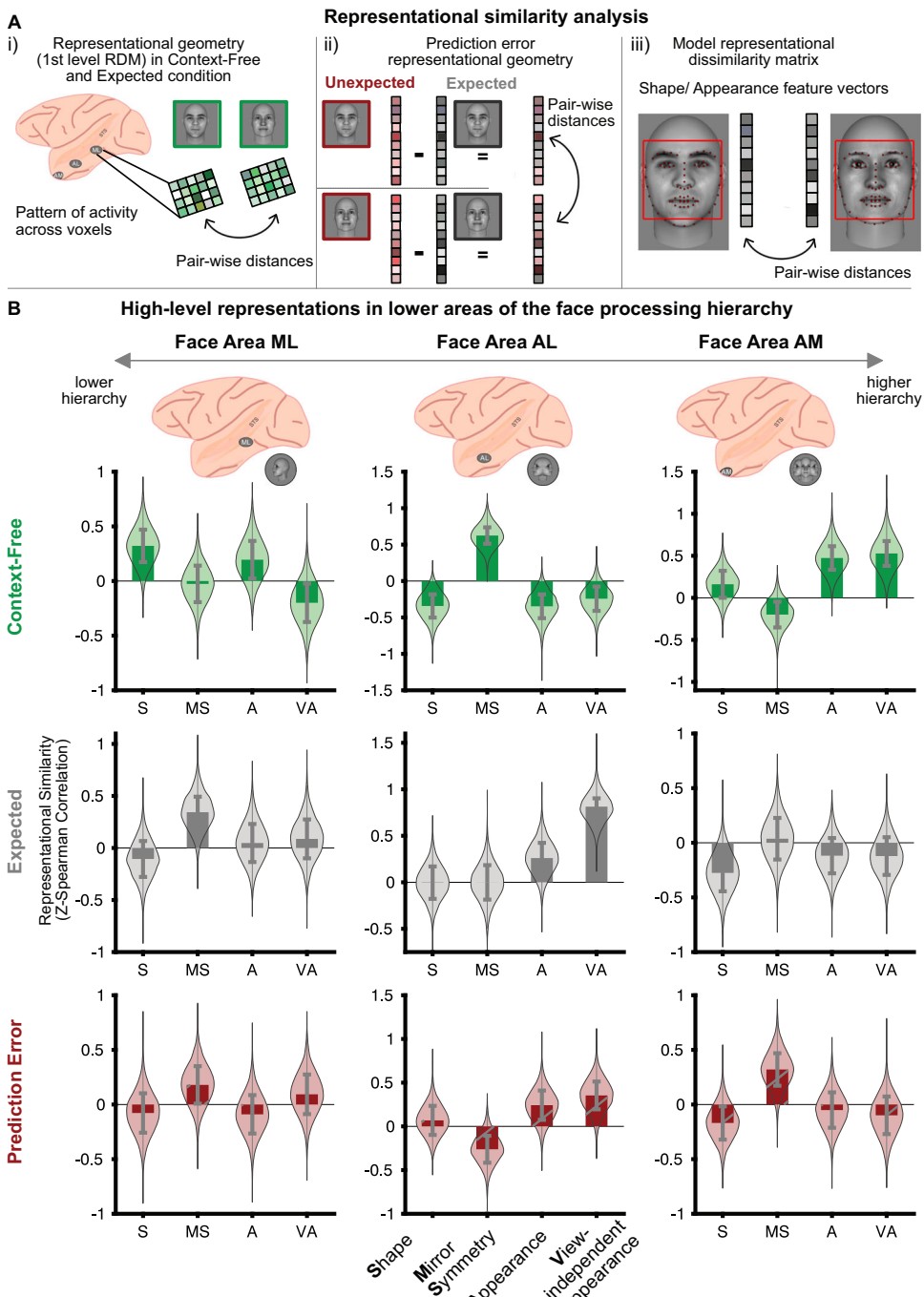

**Fig. 3 | High-level predictions cascade down the face-processing hierarchy. Ai.** Representational Similarity Analysis (RSA) was performed for each face area. 1st level representational dissimilarity matrices (RDMs) were computed for the context-free (green) and expected (gray) condition as cosine distances between the stimulus-evoked activity patterns to the different faces shown within each condition. **A.ii** Prediction error (PE) geometry was calculated on the relative activity difference of the same faces in the expected vs. unexpected condition. **A.iii** 1st level RDMs from the expected condition and PE geometry were compared to model RDMs derived from known tuning properties of the face areas (**s**hape, **m**irror-**s**ymmetry, **a**ppearance, **v**iew-independent **a**ppearance). Model RDMs were computed on distances between features vectors of the face stimuli. **B** In the context-free condition (green), activity patterns recapitulated known tuning properties of all three face areas (ML: shape vs. 3 others, $p = 0.0417$; AL: mirror-symmetry vs. 3 others, $p < 0.0001$; AM: 2 appearance-based models vs. 2 others, $p = 0.0019$, one-sided bootstrap test). When faces were expected (gray), lower areas showed increases in higher-order tuning properties compared to the context-free condition

(one-sided Raghunathan test, ML: mirror-symmetry, $p = 0.0460$; AL: appearance $p = 0.0072$ and view-independent appearance, $p = 8.941e{-}6$). This was accompanied by a decrease of feedforward tuning properties in all areas (ML: shape, $p = 0.0323$; AL: mirror-symmetry, $p = 0.0073$; AM: appearance $p = 0.0091$, view-independent appearance, $p = 0.0043$). Across areas, PE (red) geometry was highly correlated to the geometry in the expected condition (Spearman's rho $= 0.7273$, $p = 0.01$). Compared to the context-free condition, PEs in AL showed increased appearance ($p = 0.0086$) and view-independent appearance tuning ($p = 0.0079$), along with a decrease in mirror-symmetry ($p = 0.0022$). Similarly, AM showed a decrease in appearance ($p = 0.0178$) and view-independent appearance tuning ($p = 0.0049$). Violin plots show the distribution of the bootstrapped Z-transformed 2nd level similarity values ($n = 10,000$ bootstraps). Error bars depict the bootstrapped SEM of the Z-transformed 2nd level similarity. See Supplementary Table 2 for individual model fits. Source data are provided as a Source data file. The face images in panels **A** and **B** were created using FaceGen Modeller.

can change on short timescales, e.g., from spontaneous to stimulus-evoked activity[37], or with task engagement[38].

Furthermore, we find that in the presence of expectations, lower areas in the face-processing hierarchy express higher-order representations and enhanced functional connectivity with higher areas. This suggests that predictions originating from high-level face-processing areas are propagated down to lower areas via feedback connections, in line with predictive processing theories[39]. Predictions lead to a complete transformation of representational geometry, suggesting that they impose the high-level representational format characteristic of the stage of processing at which they are generated, at the expense of the feedforward tuning properties of the respective lower area. This transformation spans the entire face-processing hierarchy: ML inherits tuning properties of AL, and AL inherits tuning properties of AM. This results in the processing of facial appearance, a dimension most closely related to face identification[40], at earlier stages of processing than when no context is available. AM itself no longer shows image-bound properties expressed in the ventral face-processing network (i.e., appearance), but may inherit yet more abstract, perhaps conceptual representations from the frontal or medial temporal lobe (MTL)[41,42] with which it is intricately connected. This radical transformation between predicted and context-free representations underscores IT's capacity to flexibly and dynamically switch coding regimes on very short timescales, i.e., trial-by-trial, and refutes the notion that tuning properties are hard-wired after years of learning and development.

Whether predictions transform representations in the *entire* population of lower areas towards higher-order properties is an interesting question that we cannot answer with the spatio-temporal resolution of fMRI and therefore requires further electrophysiological studies. While a transformation of the entire population is possible, it is also thinkable that subpopulations maintain their inherent representations, while others switch[43] towards higher-order properties. Furthermore, it is also possible that transformations of representations unfold dynamically in time: the predictor face may activate top-down processes in higher face areas that subsequently act upon the successor face via late-occurring feedback signals. In line with this, our previous electrophysiological results on PEs in the face area ML[8] suggest at least two temporal phases of information passing during the successor face: an early phase with a latency of about 130 ms in which prediction errors are computed, and a later phase where high-level tuning properties like mirror-symmetry become apparent in ML. It would be interesting for future studies to disentangle the different computational components that are likely at work within a population of neurons that is thought to be minimally composed of PE and prediction units[44].

Signaling predictions and deviations from them not only improves sensory encoding but also refines internal models by means of PEs. We find that unexpected stimuli boost pattern-separation in lower face-processing areas beyond the separability found in the expected condition, while not modulating separability at the top of the hierarchy. PEs may refine internal models by increasing separability, in line with memory updating in the MTL[45] and episodic memory enhancement[46,47]. Predictive processing theories[14] suggest that PEs allow for an optimization of the available dynamic range of neurons to signal differences from the predicted face, such that several signal levels can be effectively expressed making them more separable than, e.g., the images without predictive context or in the expected condition where no PEs are computed. Furthermore, the concomitant increase in dimensionality that we find in the unexpected condition suggests that contextual information provided by predictions introduces additional dimensions into the representational geometry for separability. This would be in line with conjunctive representations between stimulus features and context[19,48], specifically between stimulus features and PEs[8] observed electrophysiologically in IT cortex.

Such mixed selectivity could boost pattern-separability and enable more effective readout from high dimensional representational spaces[28].

Previous studies have explained predictions by dampening, i.e., gain modulation[49,50], and/or sharpening of highly selective neurons[51]. While we find differences in amplitudes (i.e., lower amplitudes for expected than unexpected faces, Supplementary Fig. S3) in line with previous studies, they do not explain the differences in separability that we find across the face-processing hierarchy and between conditions, which arise from the multivariate spatial pattern of activity. Sharpening could also predict better discriminability of facial identities as a function of the underlying selectivity; however, sharpening accounts would suggest the highest increase in separability in AM because this area contains the highest fraction of identity-selective neurons[6]. Furthermore, sharpening has been argued to be specific to early visual cortex and/or low dimensional stimuli such as gratings[52,53] (but see refs. 9,50). Since neither sharpening nor dampening effects are mutually exclusive to pattern separation, unraveling the relationship of these mechanisms remains an important target for future research. Additionally, it is possible that there are differences in amplitude between predictor and successor images due to the experimental design (see Methods). However, such differential adaptation effects could not plausibly lead to inheritance of higher-order tuning properties by lower face areas.

How statistical learning effects on sensory processing are related to episodic memory remains an interesting avenue for future research. A previous study in humans investigating competitive dynamics between statistical learning and episodic memory reported that predictor images are remembered worse than successor images[54]. This might suggest that predictor images in our study were less separable than (un)expected successors by virtue of being predictive (and not because they were processed in the absence of a predictive context). However, in the abovementioned study, decoding of visual information from category-selective cortex did not predict memory performance; furthermore, a subsequent study using the same paradigm and intracranial recordings in epilepsy patients found that although some predictor images in a pair are subsequently forgotten, category information during the presentation of later forgotten and later remembered predictors is identical, and hence, that variance in memory is not due to the strength of perceptual processing of the stimuli[55]. Thus, although the relationship between predictiveness and memory warrants further investigation, separability in visual cortex in the context-free condition is likely to result from sensory processing alone.

Overall, our results provide insight into the question how the ventral stream implements predictions: top-down signals that dynamically transform neural representations into separable and high-dimensional neural geometries. Passing tuning properties down to earlier processing stages allows extraction of high-level, abstract information earlier than what is possible in the absence of predictions, when feedforward computation dominates and signals need to ascend the entire hierarchy to provide information about facial identity. Predictions thus also free resources for the processing of yet more abstract, less image-bound information in the areas whose feedforward tuning properties are now shifted upstream. Predicting on the basis of invariant tuning properties that abstract over irrelevant features circumvents overfitting to individual images and has an extended temporal horizon compared to pixel-level predictions which are prone to fail in noisy conditions. Our results thus exemplify how feedback connections enable flexible representational spaces that optimally use the computational resources provided by cortical processing hierarchies.

## Methods
### Monkey subjects and surgical procedures
Data for this study were acquired in two adult macaque monkeys (*Macaca mulatta*): one male (Monkey P, 6.5–7 kg, 9 years) and one

female (Monkey L, 8.5–9 kg, 10 years). No power calculation was performed to determine the sample size, but was chosen to be in accordance with previous studies[56,57]. We did not have sufficient sample size to investigate differences between female and male macaque monkeys. All procedures with Monkey P were conducted at the German Primate Center, Göttingen and approved by the responsible regional government office (Niedersächsisches Landesamt für Verbraucherschutz und Lebensmittelsicherheit - LAVES). The animal was either housed in pairs or groups, adhering to German and European regulations. The facility provided a stimulating environment for the animals, including various toys and structures, natural and artificial lighting, and access to outdoor spaces, surpassing the space requirements outlined in European regulations. All procedures with Monkey L complied with the National Institutes of Health Guide for Care and Use of Laboratory Animals and were approved by the local Institutional Animal Care and Use Committees of The Rockefeller University (protocol number 21104-H USDA) and Weill Cornell Medicine (protocol number 2010-0029), where magnetic resonance imaging (MRI) was performed. The animal was housed with other macaque monkeys and had access to a stimulating environment including various toys for enrichment. The well-being of both the animals, both psychologically and medically, was closely monitored on a daily basis by veterinarians, animal facility personnel, and scientists. Both animals were subjected to surgical procedures where MRI-compatible cranial head-posts were implanted, embedded within bone cement, anchored with MR-compatible ceramic screws under general anesthesia and sterile conditions[8]. The animals underwent extensive training using positive reinforcement[58] to become accustomed to entering and remaining seated in a horizontal primate chair. Their head position was stabilized using the implanted head-posts, allowing for cleaning of the implants, accurate recording of gaze/pupil diameter, and minimizing motion artifacts during MRI experiments.

We conducted this study in non-human primates using fMRI such that we can test hypotheses about pattern separation and predictive processing by imaging the entire face-processing hierarchy. Studying this in non-human primates has distinct advantages: 1. It allows direct comparison to the representation known from ground-truth electrophysiology data[6,7]. 2. Anatomical connectivity about direct feedforward and feedback connections between face areas[42] along with functional connectivity[41] has been mapped out; 3. A clear hierarchical organization between the face areas has been determined[6,59]. Such ground-truth data on connectivity, representations, and hierarchy are not available in humans, even though a face-processing network comprising areas in the occipital lobe, fusiform gyrus, and anterior temporal lobe have been found[60].

## MRI data acquisition
MRI data were acquired on 3T scanners (Monkey P: MAGNETOM-Prisma; Monkey L: MAGNETOM-Prisma_fit, Siemens Healthineers, Erlangen, Germany). Anatomical images were obtained for each animal in a separate session using a T1-weighted magnetization-prepared rapid gradient echo (MPRAGE) sequence (Monkey P: field of view [FOV] 128 mm, voxel size = $0.5 \times 0.5 \times 0.5$ mm, repetition time [TR] = 2.7 s, echo time [TE] = 2.96 ms, inversion time [TI] = 850 ms, bandwidth [BW] = 220 Hz/Px, flip angle [FA] = 8 degrees, 240 slices, 11 cm loop coil, Siemens Healthineers; Monkey L: FOV = 128 mm, voxel size = $0.5 \times 0.5 \times 0.5$ mm, TR = 2.7 s, TE = 2.99 ms, TI = 868 ms, BW = 230 Hz/Px, FA = 9 degrees, 240 slices, custom 1-channel receive coil L. Wald, MGH Martinos Center for Biomedical Imaging) while the monkeys were anesthetized (isoflurane 1.5%–2%) and positioned in an MR-compatible stereotactic frame (Kopf Instruments). Functional images were acquired using custom 4 or 8 channel phased-array receive surface coils with a horizontally oriented single loop transmit coil (H. Kolster, Windmiller Kolster Scientific, and L. Wald, MGH Martinos Center for Biomedical Imaging) while the monkeys were in a sphinx

position. Each functional time series consisted of whole-brain gradient-echo planar images (EPI; FOV = 96 mm, voxel size = $1.2 \times 1.2 \times 1.2$ mm, TR = 2 s, TE = 27 ms, BW = 1302 Hz/Px, echo spacing [ESP] = 0.93 ms, FA = 76 degrees, 43 slices) acquired in interleaved order with two times generalized autocalibrating partially parallel acquisitions (GRAPPA) acceleration. All functional images in the Statistical Learning Paradigm (see below) for both monkeys and Face Localizer (see below) in Monkey P were acquired with the above sequence parameters. Functional images for Face Localizer in Monkey L were acquired with the contrast agent ferumoxytol (Molday ION, BioPAL, Worcester, USA), and hence slightly different sequence parameters (FOV = 96 mm, voxel size = $1.2 \times 1.2 \times 1.2$ mm, TR = 2.25 s, TE = 17 ms, BW = 1525 Hz/Px, FA = 79 degrees, 45 slices). Right before every scanning session, ferumoxytol was injected into the saphenous vein. The amount administered ranged from 9 mg of Fe per kg of the animal's body weight during the first scan to 6 mg on following days to account for the functional half-life of the contrast agent. On every session, field maps (FOV = 96 mm, voxel size = $1.2 \times 1.2 \times 2.4$ mm, TR = 0.7 s, TE1 = 5 ms, TE2 = 7.46 ms, FA = 60 degrees, 22 slices) were also acquired which were later used for EPI undistortion[61].

## Face localizer
We use a standard face localizer to localize the face patches ML, AL, and AM[62]. Subjects fixated on a central white dot while we displayed images of human/monkey faces, body parts/headless bodies, man-made objects, and fruits, alternating with baseline periods in a block design (FOB). Each block lasted 24–30 s. Fluid reward was given after variable 2–4 s periods, contingent on gaze staying within 2 degrees of the fixation dot. Analysis included runs where subjects achieved ≥80–85% fixation stability. Visual stimulation and reward were controlled using Psychtoolbox[63] for Monkey P and Presentation (v19, Neurobehavioral Systems) for Monkey L. Stimuli were projected on a back-projection screen using a video projector (Monkey P: Epson EB-G5600, refresh rate 60 Hz, resolution 1920 × 1080 px; Monkey L: NEC NP3250, refresh rate 60 Hz, resolution 1024 × 768 px) with a custom lens. Eye position was measured using a video-based eye tracking system (Monkey P: SR Research Eyelink 1000 NHP Long Range Optics at 1000 Hz, Monkey L: ISCAN at 120 Hz).

## Statistical learning paradigm
For the statistical learning experiment, we generated 30 three-dimensional faces with neutral expression and no hair using FaceGen Modeller Pro 3.27 (Singular Inversions). Seven views (0°, 30°, 45°, 60°, 300°, 315°, 330°) were extracted for each face. The faces were counterbalanced for their gender and skin texture was added such that the right profile image was not a simple mirror-version of the left-profile image. The images were converted to grayscale and then luminance normalized using the SHINE toolbox[64]. The training set for the statistical learning experiment consisted of 18 facial identities with 6 images of frontal, right and left profile-view each. These were split into 9 predictor-successor pairs such that the head orientations were balanced within predictors and successors, respectively[8]. There are no separability differences between the set of predictor and successor face-stimuli in terms of low-level image properties (gabor-filterbank, t(70) = −0.3525; p = 0.7255) as much as in terms of shape (t(70) = 0.6015; p = 0.5494) and appearance (t(70) = −0.9056; p = 0.3682).

Prior to the training phase, monkeys were familiarized with the entire set of 30 faces in all the 7 views in randomized order such that no image was novel to the monkey. The monkeys were then exposed to the face-pairs to establish associations between the predictor and the successor images. Pairs were arranged such that one identity-view combination would uniquely predict one other identity-view combination (Supplementary Fig. S4). Training was conducted with sequentially presented face-pairs for at least 3 weeks for both monkeys. The sequence of pairs was designed such that the transitional

probability within a pair was 100% and the transitional probability between pairs was kept at a minimum.

In the early phase of training, each image was presented foveally (8 dva in size) for 500 ms with an inter-stimulus interval of 500 ms. Longer baselines (up to 9.5 s) and spatial jittering (-1 dva) from the foveal position were gradually introduced over the training period. This was done to accustom the animals to the task design to be used during the fMRI experiments. The monkey was rewarded with water/juice every 2.5–3.8 s for maintaining fixation within a 2–3 dva window around the fixation dot presented at the center of the screen. To control visual stimulation and reward delivery, we used mWorks (https://mworks.github.io/) and a custom-built input-output device (R. Brockhausen) using Teensy 3.6 (PJRC: Electronic Projects) for Monkey P, and Presentation (v20.2, Neurobehavioral Systems) with a commercially available data acquisition device (Measurement Computing USB 1208FS) for Monkey L. Stimuli were presented using a video projector for Monkey P (Barco F22 WUXGA; refresh rate, 60 Hz; resolution, 1920 × 1080 px) and displayed on a monitor for Monkey L (Dell E2011H, refresh rate 60 Hz, resolution 1920 × 1080 px). Eye position was measured using video-based eye tracking (Monkey P: SR Research Eyelink 1000 at 1000 Hz, Monkey L: ISCAN at 120 Hz).

The testing phase of the statistical learning paradigm took place while fMRI data were acquired (Monkey L: 81 runs; Monkey P: 56 runs). In this phase, learned face-pairs were presented such that one face predicted the next; violations of the predictions (40% of all trials) were introduced in the second image of a face-pair. The violations entailed a successor image with a different identity (but with the expected view) or a different view (but with the expected identity) than during the training phase. To create the identity violation conditions, we systematically recombined the trained predictor image of a pair with that of another successor image to generate unexpected pairs. This was done to avoid novelty effects, such that unexpected identities were not novel identities for the monkey. To create the view violation conditions, we presented the predicted identity but with a different head orientation than during training. The experimenter was not blinded to the experimental conditions. A fixation dot was continuously presented at the center of the screen and the monkey was rewarded for maintaining fixation within a 2–3 dva fixation window every 2.2–3.6 s. Reward was thus not contingent upon image presentation but entirely on fixation performance. We used a rapid-event related design: each face image was presented for 500 ms, aligned to the onset of volume acquisition. To facilitate single-trial deconvolution of the blood oxygenation level dependent (BOLD) response, baseline durations were jittered between 1.5, 3.5 and 5.5 s for intra-pair and 5.5, 7.5 and 9.5 s for inter-pair intervals. The face images were spatially jittered within -1 dva around the fixation dot to avoid low-level adaptation effects.

## MRI processing

Anatomical images were intensity normalized, skull-stripped, and segmented using Freesurfer (v5.3, https://surfer.nmr.mgh.harvard.edu/)[65]. Preprocessing of fMRI data was done using Freesurfer's functional analysis stream, FS FAST, AFNI[66] and the JIP toolkit (https://www.nitrc.org/projects/jip). Slice-wise motion-correction was done within each run using AFNI's 3dAllineate with terms for cubic warping in the phase encoding direction, as well as shifts, rotations, scaling, and skewing, followed by slice-time correction using FS FAST. Geometric distortions of the functional volumes were corrected by means of the field maps, followed by mutual information-based non-linear alignment to the high-resolution anatomical scans as implemented in JIP.

Statistical modeling was done using a General Linear Model (GLM) in FS FAST. Predictors of interest were convolved with the canonical hemodynamic response function (HRF) for BOLD for all experiments except for Monkey L's Face Localizer data where HRF parameters were appropriate for ferumuxytol[67]. Motion parameters, fixation stability, and reward were included as nuisance regressors. The first five volumes of each functional run were excluded to prevent T1 saturation artifacts, and detrending (up to 2nd degree polynomials) was applied within the GLM. To identify the face-areas from the FOB data, we contrasted faces vs. all other visual categories[59]. Regions of interest (ROIs) in each hemisphere were defined as the intersection of a sphere around the peak voxel identified by this contrast and the gray matter, resulting in ROIs of -100 voxels. Since face-processing is not lateralized in macaque monkeys[68], we combined the voxels in each of the face-areas from the two hemispheres for all our analyses (Monkey P: ML 204, AL 185, AM 187; Monkey L: ML 198, AL 190, AM 204 voxels).

For the testing phase of the statistical learning experiment, we resolved trial-specific activation using the least squares-single approach[69]. In this approach, a separate GLM is calculated for each trial and the design matrix contains two main task-related regressors: the trial-of-interest was defined as the first regressor and all other trials in a run, irrespective of the experimental conditions, were simultaneously defined as the second (nuisance) regressor. This was done iteratively for every stimulus within a run. The regression coefficient for the trial of interest was contrasted with the baseline and the t-statistic obtained for each trial was used for further analyses[70].

## Pattern separability

Pattern-separability was computed across facial identities using cosine distances, a measure commonly used in the pattern separation literature. Cosine distances indicate the vector direction of a population response irrespective of amplitude or sample size differences in the experimental conditions[71–74] and have thus been argued to be particularly sensitive to the tuning of neural populations in the context of fMRI[75]. To compute cosine distances, we extracted t-statistics from all trials and conducted multivariate noise-normalization using the covariance matrix of the residuals from the GLM done on every trial[70]. Cosine distances were then converted to angles by computing the inverse cosine. This was done for each stimulus in all ROIs and conditions, respectively. All nine facial identities that were presented as the first stimulus in a pair were used to estimate the separability in the 'Context-Free' condition. The first image in a pair cannot be anticipated on the basis of the preceding image and is thus processed in the absence of contextual predictions. All nine trained successor stimuli were used for the 'Expected' condition. The distances between the same nine stimuli when presented as an expectation violation were used to quantify the separability in the 'Unexpected' condition. Due to the circular nature of angles, circular statistics[76] were used to compare conditions and ROIs, respectively. We used the Hotelling paired sample test for equal angular means[77] for paired comparisons, and the William-Watson test for independent samples comparisons[78]. All p-values were corrected for multiple comparisons using the Bonferroni–Holm method[79]. All the above analyses were done using The Decoding Toolbox[80] in MATLAB (The MathWorks, Inc., R2018B) and custom MATLAB scripts.

## Dimensionality

To estimate dimensionality of neural representational space in a given ROI in the different experimental conditions, we computed the Participation Ratio (PR)[15,16,81]. PR is the ratio of the square of the first moment and the second moment of the eigenvalue probability density function (PR = $(\sum_i \lambda_i)^2 / \sum_i \lambda_i^2$, where $\lambda_i$ are the eigenvalues of the covariance matrix). It quantifies how evenly variance of activity is spread across stimuli; i.e., if variance is widely spread, PR is high. We estimated PR using trial-averaged, whitened t-statistics across voxels and stimuli in each condition and ROI. Hence, our measure of dimensionality is constrained by the number of stimuli. To determine whether the dimensionality estimates are interpretable, we compared the estimated dimensionality to that of noise. Noise ceilings were estimated by synchronized permutation (1000) of the stimuli and voxels for each condition and ROI. We then tested whether the estimated dimensionality is lower than the 95% confidence interval (CI) of the noise

distribution. Similarly, pairwise differences between the dimensionality across ROIs or conditions were assessed by comparing whether the increase in estimated dimensionality was higher than the 95% CI of the noise distribution. All *p*-values were corrected for multiple comparisons using the Bonferroni–Holm method[79]. PR was quantified using publicly available[45] Python scripts and statistically analyzed using custom MATLAB scripts.

## Population response magnitude

Population response magnitudes[19] were quantified as the Euclidean (L2) norm across all voxels in a given ROI for each stimulus separately and then averaged. This served to disentangle differences in the magnitude of the IT population response from angular differences in stimulus identity as it has been shown that population response magnitude and population response vector direction can contain differential information[82]. We computed the Euclidean (L2) norm from the t-statistics per stimulus without further normalization so as not to discard any relevant magnitude differences, as recommended in ref. 19. Paired t-tests were used to assess differences in population response magnitude across expected and unexpected conditions, and corrected for multiple comparisons using the Bonferroni–Holm method[79].

## Representational similarity analysis

Representational Similarity Analysis (RSA)[11] was conducted to assess tuning properties of face-areas and how they change with experimental conditions. We first determined whether we could replicate the well-known tuning properties of the face-areas using BOLD signals, relative to the electrophysiological "gold standard"[6,7]. To this end, we computed 1st level representational dissimilarity matrices (RDMs) using cosine distances (as described in the separability section above) on the pattern of BOLD responses across voxels in each of the ROIs. We then compared these 1st level RDMs to representations predicted by previous electrophysiology studies (Model/Hypothetical RDMs).

The choice of model RDMs was based on previous studies that show that ML has a view-specific representation of facial shape, AL has mirror-symmetric representation, and AM has view-invariant representation of the facial appearance[6,7]. To compute hypothetical RDMs for shape and appearance, the Active Appearance Model[83,84] (https://www.menpo.org/) was trained using 871 pre-annotated images and used to fit each of our face stimuli to obtain 68 shape landmarks[85]. We then computed Spearman rank correlation distances between the feature vector of landmark positions for all the stimuli to compute a hypothetical shape RDM. We used 45° and 135° faces to extract all 68 shape landmarks to ascertain a full set of shape landmarks for all the facial views such that similarity between the faces could be computed considering all views. Next, to compute an appearance representational space, each face was smoothly warped to the average shape template made from the training set, using spline interpolation. This warped image was then normalized for the mean and reshaped to obtain an appearance feature vector. We then computed Spearman rank correlation distances between the feature vector of the appearance feature vector for all the stimuli to obtain a hypothetical appearance RDM. Note that this appearance model RDM contains a component of facial view and is different from the shape-free appearance model referred to in ref. 7 where the authors computed appearance representations on each view separately. Therefore, we additionally computed a model RDM using only the frontal-view versions of the same nine facial identities and performed the same procedure as above to obtain the hypothetical appearance RDM. Although this hypothetical RDM is made with only frontal faces, correlating it with an empirical RDM based on three different views is what makes this a view-independent/invariant appearance model. Lastly, we designed a hypothetical mirror-symmetry RDM by assigning the distance within each view and between left and right profile faces to be zero[6].

Comparisons of model RDMs with empirical, 1st level RDMs were carried out using Spearman rank correlation[11], partialling out low-level similarities[86] using a Gabor wavelet pyramid as a model of early visual cortex[87,88] (https://github.com/daseibert/image_similarity_toolbox). To test whether the known electrophysiological properties of each face-area best explain the representations in each of the three face-areas, we compared the model fit of the candidate RDM (e.g., Fisher *z*-transformed Spearman rank correlation between Mirror-symmetry model RDM and the 1st level RDM in AL) to the average model fits of the three other models (e.g., shape, appearance, and view-independent appearance). This was done using bootstrapping (10,000 samples, one-sided) in the context-free condition when no predictability exists, and which corresponds most closely to the electrophysiological studies that established the tuning properties of the face-areas. To test whether tuning properties changed as a function of experimental condition, i.e., whether there is an increase in higher-order tuning properties in lower areas and a decrease in feedforward tuning properties in higher areas, respectively, we compared the model fit of the candidate RDM in the condition with no context to that of its model fit in the expected/unexpected identity conditions. This was done by using Raghunathan's test[89] (one-sided) for dependent, non-overlapping correlation coefficients which also takes inter-correlations between the model and experimental RDMs into account.

To study the representational geometry of PEs, we first computed the relative difference between the expected and the respective unexpected condition (view, identity). This effectively isolates PEs, which are thought to signal the difference between expected and unexpected stimuli with increases in neural activity[8]. Following this, we computed pairwise cosine distances between the change in activity pattern and determined 2nd level similarity by means of RSA[27]. For the unexpected identity condition, cosine distances were computed across the different identities before the 2nd level RSA was conducted. For the unexpected view condition, we calculated the difference between the expected identity and the view violations of the corresponding identity (both unexpected views of a given identity separately). Then, the 1st level RDM was computed across the different face images, followed by 2nd level RSA (the Gabor wavelet pyramid was not partialled out for this analysis). Since PEs are thought to be computed on the representational format of expectations, we hypothesized that PEs also lead to an increase in high-level properties in lower areas of the hierarchy. We assessed increases in high-level tuning properties and decrease in feedforward tuning properties between unexpected identity and context-free conditions using Raghunathan's test for dependent, non-overlapping correlations[89]. To compare the context-free with the unexpected view condition, we used Fisher's test for independent, non-overlapping correlations[90]. These analyses were carried out in R (v3.6.2) using the 'cocor' toolbox[91].

## Pattern connectivity

We conducted pattern connectivity analyses[11] to investigate how much shared information exists between pairs of face areas. For each condition, pattern connectivity was conducted as a repeated measures correlation[92] – using 1st level RDMs while the variance across the repeats, i.e., representations of each brain-area pair within the face-processing hierarchy was taken into account. Additionally, we controlled for spurious pattern connectivity by partialling out low-level similarities[86] using the Gabor wavelet pyramid[87,88]. Finally, we tested whether there is an increase in pattern connectivity in the expected condition as compared to the context-free condition using a bootstrapped t-test.

## Pupil entrainment

Pupil entrainment was used to determine whether the face-pair structure was learned, in accordance with previous studies establishing neural[55,93] and pupillary entrainment[10] as a signature of statistical

learning. During training, face images were presented at a rate of 1 Hz (the image frequency), resulting in a pair frequency of 0.5 Hz. To determine pupil entrainment to face pairs, we computed inter-trial phase coherence[94] (ITC). Only runs in which fixation stability was >85 % for Monkey P and >80% for Monkey L were selected. Preprocessing of pupil area data was done in accordance with previously published work on pupil entrainment to face-pairs[10]: blinks and missing data were linearly interpolated; outliers in pupil area were detected using median absolute deviation (cutoff 3.5 for Monkey P and 1.5 for Monkey L) and linearly interpolated. This was followed by low-pass filtering at 5 Hz using a onepass-zerophase Kaiser-windowed sinc finite impulse response (FIR) filter (filter order 1812, transition width 2.0 Hz, pass band 0–4.0 Hz, stop band 6.0–500 Hz, maximal pass band deviation 0.0010 (0.10%), stop band attenuation −60 dB)[95]. Subsequently, data were detrended per run and per block; the average pupil area over the baseline of 2 s before the beginning of stimulation in each block was subtracted. Data from Monkey P were then downsampled to 500 Hz and Monkey L was retained at the acquisition frequency (120 Hz). Pseudo-trials of 32 seconds were made from continuous data, time-locked to the first stimulus in a face-pair. The pseudo trials had 30 second overlap for Monkey P and 26 second overlap for Monkey L. An additional baseline correction was performed per pseudo-trial by subtracting an average of the pupil area over the 2 s pre-block baseline. Data for Monkey L which was acquired at a low sampling rate was additionally detrended per pseudo-trial. We then computed a trial-by-trial Fourier transform using discrete prolate spheroidal sequences (DPSS) as tapers with a spectral resolution of 0.0625 Hz, using the Matlab toolbox Fieldtrip (v20190329; http://www.fieldtriptoolbox.org/)[96]. The resulting complex spectra were used to calculate ITC. ITC at the pair-frequency was normalized by dividing the mean ITC of 4 surrounding frequency bins (2 above, 2 below). Normalized ITC values above 1 thus indicate entrainment at the respective frequency. Statistical significance of ITC at the pair-frequency was determined using a one-sided t-test against 1 in both monkeys. In addition, to assess whether ITC at the pair frequency increased from early to late phases of training, we compared the first and second half of training sessions, respectively, in monkey P (unpaired, one-sided t-test). Due to technical issues and time constraints, pupil data for monkey L was only available from the late training phase.

### Reporting summary

Further information on research design is available in the Nature Portfolio Reporting Summary linked to this article.

## Data availability

Source data are provided with this paper. The data for the figures in this study have been deposited in the Figshare database under accession code (https://doi.org/10.6084/m9.figshare.24233185). Further information and requests for resources should be directed and will be fulfilled by Caspar M. Schwiedrzik (c.schwiedzik@eni-g.de) upon request. Source data are provided with this paper.

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

## Acknowledgements

We thank W. Freiwald for generously allowing us to collect part of the data in his lab; I. Kagan for support; R. Auksztulewicz for insightful discussions; L. Melloni for comments on the manuscript; A. Gonzalez, L. Burchardt, R. Mielsch, and S. Plewe for help with animal training and care; T. Becker, O. Batura, and A. Schrod for veterinary care; S. Karmacharya, W. Zarco, and R. Ludwig for help with the respective setups; and L. Yin for administrative support; Mariia Shitik (M.S.) for making the graphics (of faces) used in Fig. 1A. This research was supported by the Deutsche Forschungsgemeinschaft (DFG, German Research Foundation) through an Emmy Noether grant (SCHW1683/2-1) and SFB 1528 - 'Cognition of Interaction' (Project-ID 454648639, project A04) to C.M.S.; additional support was provided by an Outgoing Grant from the Leibniz ScienceCampus 'Primate Cognition', Göttingen, Germany to T.N. We further acknowledge support by the Open Access Publication Funds/ transformative agreements of the Göttingen University.

## Author contributions

T.N.: Conceptualization; Methodology; Software; Validation; Formal analysis; Investigation; Writing - Original Draft; Writing - Review & Editing; Visualization. C.M.S.: Conceptualization; Methodology; Writing - Original Draft; Writing - Review & Editing; Supervision; Project administration; Funding acquisition.

## Funding

## Competing interests

The authors declare no competing interests.
