## [Peer Review File · Nature Communications]

Predictions enable top-down pattern separation in the macaque face-processing hierarchyReviewer #1 (Remarks to the Author):

This is a great manuscript, investigating the effect of expectation after pair learning on the encoding of information within a well-defined hierarchy of cortical areas in macaque monkeys, as measured with fMRI. The authors find that the encoding of the expected second images within pairs becomes more separable relative to the encoding of the first images within pairs. Furthermore, when the second image of a pair is unexpected, the encoding becomes even more separable. Furthermore, with an elegant analysis, they show that the features that make the encoding more separable indicates top-down processing within the hierarchy.

The design of study is creative and well thought out, the results are clearly presented, both in the figures and in the text, the methods are appropriately used, and the results are novel and highly relevant for the field.

I have only one point about the interpretation that I think could be a bit more nuanced. On lines 336-347, it is suggested that the results could also be interpreted in terms of dampening and sharpening. Some arguments against these interpretations are given, which are not all that convincing to me :

1) It is written : "Dampening predicts amplitude differences between expected and unexpected stimuli; while we find such differences in amplitudes in line with previous studies (Supplementary Fig. S2), they do not explain the differences in separability that we find across the face-processing hierarchy and between conditions, which arise from the multivariate spatial pattern of activity." However, the effect of the magnitude of the response for the different conditions (Fig. S2) seems nicely in line with the effect of separability (Fig. 2B), namely : higher for expected than 'context-free' and higher for unexpected than expected.

2) It is written : "sharpening accounts would suggest increased separability solely in AM because only those neurons are highly identity-selective."

There is some identity selectivity in ML and AL as well, as far as I know, which could then also be sharpened, right? In fact, it could be argued that sharpening could have a stronger effect in ML and AL than in AM, as the selectivity is farther from the noise ceiling?

3) It is written that : "sharpening may be specific to early visual cortex and/or low dimensional stimuli such as gratings"

This does not seem correct. In fact, initial work on sharpening of responses in monkey cortex showed the effects for natural objects in IT, see the work of Travis Meyer and Carl Olson, in particular : <http://www.nature.com/doi/10.1038/nn.3794>

In general, I would say that the finding of the increase separability is complementary to the ideas of dampening and sharpening. There does not seem to be any need to choose between one and the other as far as I can see. Furthermore, the current manuscript add on top of what is known, by showing that the effect of separability indicates top-down input, which could be true for dampening and sharpening as well.

In this respect, it could be relevant to add a reference to preliminary work by Koyano et al 2023 on dampening in different macaque monkey face patches:

<https://www.abstractsonline.com/pp8/#!/10892/presentation/38672>

They find that familiarization only leads to long lasting changes in a high-level face patch (AM and not AF), which could lead to top-down modulation of these effects in lower-level face patches, if they would have been sensitive to these modulatory effects as in line the current study.

Minor points :

- It is written : "we also found stronger responses in the unexpected than in the expected condition in all three face-areas (Supplementary Fig. S2; all $p < 3.5e-04$). However, this difference in magnitude did not explain the increase in pattern-separability (Spearman's $\rho = 0.0167$, $p = 0.9816$) nor dimensionality ($\rho = 0.2667$, $p = 0.4933$)"

It is not clear how this analysis is done. I assume that the correlation is calculated between the magnitude and the pattern-separability of the different stimuli, averaged across areas and conditions? Please clarify this in the text. Currently, it's difficult to interpret this result.

- It's not clear how many times the different face identities are presented within a test block. If I understood correctly, there are 9 facial identity pairs and 12 additional unpaired facial identities? I assume that the unpaired faces are also presented during test blocks, during incongruent trials? If

so, this might explain part of the effects, due to stimulus-specific adaptation, or dampening as the authors write. For all trials, the 'context free' condition are using one of only 9 faces. The amplitude of the response of these should then be lowest, as adaptation is strongest, and particularly for AM where identity selectivity is highest, which would be in line with the results in Fig. S2. Then, 60% of the trials are congruent, so the faces used for the expected condition appear more often than the faces used for the unexpected condition (assuming that there is a difference between the sets of faces used for these two conditions). This could again explain the effects shown in Fig. S2, where the effect of adaptation could be stronger for the expected condition than the unexpected condition. If this analysis is correct, it would be useful to add it to the discussion. In any case, it would be good to be more explicit about which facial identities are used for the different conditions, and how many times a facial identity is presented.

- I don't manage to find how much data was recorded. How many blocks per monkey? How many trials per block?

- It is not clear from the title, abstract nor figures that this is an fMRI study. I think it would be useful to make that explicit in the abstract, or in the first or second figure. I would also have expected to see some results at the whole brain level, at least the results for the localizer for both monkeys, to convince the reader that the different face patches could be reliably located in both subjects.

Reviewer #2 Attachment on the following page

The authors study how contextual predictions influence the neural code in earlier visual regions during face recognition. Using fMRI in monkeys and focusing on the ventral visual system, the authors find that items that were expected or unexpected produced separable neural patterns in earlier areas that do not show such neural codes typically. This is an interesting study with clear results that advance our knowledge about predictive processing in the brain. I have some concerns about the clarity of the logic and some of the analyses.

Major concerns:

1. I am not sure that the predictive items should be interpreted as baseline or context-free. Indeed, these items are not predicted. However, they do have predictive content – i.e., they predict the following items. This predictiveness has previously been shown to influence learning and memory as well as neural representations of items (e.g., Sherman et al., 2020, PNAS). In fact, one possibility is that predictive items, previously shown to be remembered at lower rates, might have low separability of their neural representations. Thus, it could be that rather than the predictability of items improving separation (in the Expected condition), it is the predictive cue (currently defined as baseline) that has lower than baseline separation. Another potentially preferable baseline might have been items that were interspersed between the pairs randomly, such that they are as familiar as other items, but are not predicted nor do not predict the following item. The question of an appropriate baseline is a difficult one, which might not have a straightforward answer, and it is impossible to change in the current experiment. I would recommend changing the conceptualization of the ‘baseline’ to a ‘predictive’ condition and modifying the interpretation of the results accordingly. To be clear, I still believe the results are important and interesting even under this interpretation.
2. Why was the condition of inducing prediction errors using different viewpoints included? It seems to reduce power and was not part of the main hypothesis. If that is for future analyses, or for analyses that did not pan out, or if this is a reanalysis of a previous dataset, this should be clearly laid out.
3. Please provide a clearer explanation or intuition of the math behind the dimensionality analysis. Specifically, it was hard to evaluate whether this measure is independent of the cosine distance, given that both cosine distance and eigenvalues are related to the covariance matrix of the data. I am concerned that this is not an independent analysis that warrants its own interpretation, rather than a different computation that captures similar aspects of the data like the cosine distance. This concern is reinforced by the high correlation between the measures.
4. The logic of the PE-RDM analysis (reported starting in line 255) is unclear to me:
 - a. The properties captured by the RDM are the features of the faces (e.g., viewpoint, invariance etc.), not the PE signal which was computed as a difference between patterns. Why compare the difference between expected vs. unexpected, vs. just the representation of faces in the unexpected condition (as was done in the other analyses)?
 - b. It was unclear to me what this PE signal reflects. PE is usually thought of as a scalar value, potentially the difference in the average magnitude of the signal. How do the

authors interpret this multivariate PE signal? Is that saying that each voxel has its PE signal, and looking at the pattern is a way to summarize that? Please clarify.

- c. Mathematically, this analysis is not independent of the analysis in the Expected condition – because the Expected patterns are subtracted from the Unexpected. Thus, all the differences in similarity to RDMs might stem from the Expected patterns. Indeed, the patterns correlated. As currently done, I don't think this can be interpreted in a meaningful way separately from the Expected results.
5. While I agree with the authors' general claim that properties of high-level regions are apparent in low-level regions in the Expected and Unexpected conditions, I would be more cautious with interpreting the results as meaning that these features 'cascade' or are transmitted from high-level regions. Currently, I believe that there is no data directly supporting this point in the paper. These features could emerge from altered internal properties, or they can be transmitted from earlier regions, or different regions not tested (e.g., the hippocampus or the prefrontal regions that are also important for predictive processing). To give some initial support for this part of their argument, I would recommend measuring functional connectivity between regions and testing if it is related to separability or the other neural effects reported. For example, I imagine that there was some variance in the amount of separability between trials. The authors could leverage this variance and bin their trials (maybe to thirds) based on whether they are high/medium/low separability (in each experimental condition) in a lower area, and compute correlation in activity between areas (e.g., using beta-series connectivity, Rissman et al., 2004; or gPPI). The prediction is of higher connectivity in high vs. low separation. This wouldn't show directionality between regions, of course, or exclude the possibility that other regions are involved in orchestrating communication, but would provide initial support that communication between regions is related to separability.

Minor concerns:

1. Line 27: I think the sentence is nongrammatical. Should be "The reason for why we struggle... " or "The reason for struggling to distinguish..."
2. Line 35: the hypothesis that "incorporating higher-order predictive information leads to more separable (Fig. 1A) and invariant (Fig. 1B right) representations already in lower areas of the hierarchy" Is not substantiated. I understand everything that precedes that hypothesis, but it does not lead to the hypothesis. Why would that be the solution? For example, it can be that face recognition is done in such cases based on higher-order areas that have separable neural patterns. Or why would predictions from higher-order areas that are communicated to lower-order areas lead to separable codes? Is it because there are separable neural populations activated in these higher-order areas, thus they communicate, and activate separable populations in lower-order areas? As currently stated, the logic is unclear.
3. Given that the authors use a non-invasive fMRI method to study their question, I think it is worth mentioning why choosing a model animal like monkeys, rather than directly testing the question in humans (which are easier to test and have a well-characterized face processing system). Is it an intermediate step in translating electrophysiology research in monkeys to human fMRI (and if so, were the findings reported in the paper previously

reported in electrophysiology)? Or another reason? This might be self-evident for monkey researchers, but less so for the broad audience of Nature Communications.

4. Figure 1B: What are S1,S2,S3? Why not use ML, AL, AM?
5. Line 118: I'm not sure I agree that "This suggests that high separability *otherwise characteristic of the top of the processing hierarchy* can already be attained in lower areas if predictive information exists - potentially facilitating readout of information from well-decorrelated representations." The authors show an increase in separability, but regions ML and AL are still about 10-20 degrees less separable than AM. Why do the authors argue that they achieve high separability like at the top of the processing hierarchy?
6. Line 122: the logic of why unexpected faces would increase pattern separation more than expected faces is unclear to me. The authors write: "Predictive processing theories suggest that predictions serve to compute prediction errors (PE) as a deviation between the prediction and the sensory input to reduce redundancy in information transmission: redundant information is removed, optimizing the dynamic range of neurons and allowing them to efficiently signal decorrelated or unpredicted information. In terms of pattern separation, this may entail higher pattern-separability when PEs (to unexpected faces) occur." I agree that PE should reduce redundancy in information, and that could lead to separate representation. However, that should promote learning and predictability, such that the expected faces should be those for which the redundancy/noise has already been removed, and separability is highest. Please explain. I'm not arguing that there couldn't be reasons why PE would increase separability, and potentially more than expected stimuli (e.g., Frank et al., JNeuro, 2020; Kim et al., JNeuro, 2017; Bein et al., 2023), but that I don't understand the logic of the argument laid here.
7. Line 226: I'm not sure why this is the first introduction to the analysis as an RSA analysis. The cosine distance analysis performed earlier in the paper is also a type of RSA. Please consider moving that definition and ref earlier.
8. Figure 3, Aii is confusing, because it seems like a step in the analysis between Ai and Aiii, whereas, if I understood correctly, it is not (for the context-free and expected conditions). I'd recommend moving it to next to the Prediction error panel in B (or changing the analysis altogether, see major comment 4).
9. In the Results, section "high-level predictions cascade down the face-processing hierarchy", please describe briefly what is the known electrophysiological tuning property of each region (based on the figure and later in the text, I believe it is view-specificity/mirror symmetry/view invariance). It would be useful to include it already in the initial description of the electrophysiological tuning. Additionally, please describe shortly how would that translate to an RDM – e.g., if view invariance, was the RDM capturing high similarity between different viewpoints of the same face, but lower similarity between different faces?
10. Line 249: why would predictability reduce the view-independent appearance in AM? How do the authors explain that?
11. Line 536: was the multiple correction done for 3 ROIs? Or the number of comparisons?
12. Please clearly state in the Methods what was the sample for all tests. For example, in line 116, what does 70 df's reflect? Or in line 136, what do 34 df's reflect?

Reviewer #3 (Remarks to the Author):

This manuscript investigates how contextual information affects the neural representation of faces in three face areas of the macaque brain using functional MRI. After exposing the animals to temporal associations of nine pairs of faces, they found that the representation of faces depends both on the temporal order of the presentation and on whether the successor face can be predicted by the associations. Their results suggest that the responses to successor faces reflect “top-down” signals from higher-level face areas. Overall, this is a well-conducted study, with new insights into how statistical learning affects neural representation. Although there’s a lot to like in their paper, I have a hard time understanding the computational implications of the results. My specific comments are as follows:

Major:

1)The interpretation of the results. The senior author of this paper previously used this paradigm in macaques and claimed that lower-level face regions compute prediction errors based on prediction signals from higher-level face areas (Schwiedrzik and Freiwald, 2017). In this paper, the comparison between expected and unexpected stimuli can also be explained by the predictive coding framework. However, it’s unclear to me how the comparison between expected and context-free stimuli fits into the picture. There should be no error for the expected stimuli, so what’s the nature of the representation? The current narrative focuses on explaining the what the data look like, but a single coherent framework is not in sight.

2)A surprising result in Figure 3 is that while lower-level face areas inherit representation from higher-level areas, they also discard their original representation. I’m puzzled by this result because one would expect the higher-level representation to come from a feedback signal, but that representation would have to be in the higher-level area first in order to be transferred to the low-level area via feedback later on. This conflict may be resolved if the feedback signal comes only from the context-free predictor, but the feedforward signal should still be present so that the prediction error can be computed. Relatedly, in the last row of Figure 3, which faces are used to compute the similarity matrix, the successor faces in the unexpected condition? If so, the representation cannot come from the predictor face, and I would expect to observe the representation at the original level. Alternatively, the loss of the original representation format may be due to the lack of temporal resolution of the fMRI technique. In any case, this should be thoroughly discussed.

3)What is the composition of the nine face pairs? Will the same face appear as both a predictor and a successor? If not, then the face pairs being compared are not perfectly matched. This may affect the results in Figure 2. For example, the nine successor faces may be more separable from each other in the neural space than the nine predictor faces. Please clarify the procedure and estimate the size of the effect.

Minor:

1)Last line on page 4: “Pattern-separability increased along the hierarchy (ML<AL<AM)”. Where is the data shown, Figure 2B? Please refer to the figure.

2)In several cases, p-values of statistical tests were reported in the figure legends and in the main text, but detailed information, including the type of statistical test and the degree of freedom, was missing. Please provide this information.

3)Page 7, line 187. The correlation between increases in dimensionality and separability across conditions and hierarchy was computed and tested for statistical significance. Since each value here is a difference between two values that could appear in other comparisons, the samples cannot be considered independent. Is the statistical test corrected for this?

4)As pointed out by Dubois et al. 2015, information that can be decoded from single-unit responses may not be available in fMRI response patterns. This technical limitation should be discussed by the authors.

REVIEWER COMMENTS and RESPONSES

Predictions enable top-down pattern separation in the macaque face-processing hierarchy

Tarana Nigam and Caspar M. Schwiedrzik

We thank all the 3 reviewers for their positive remarks about our study, a careful assessment of the manuscript and for providing constructive feedback. We have incorporated the feedback provided by the reviewers in the manuscript, provided additional supplementary figures and have also done additional control analyses. We have provided our point-by-point answers to the questions and suggestions from the reviewer (in blue) and hope they provide enough clarification, satisfaction to the reviewers. We also thank the editor, Daniel Barry, for his prompt handling of the manuscript and his input. We have also modified the manuscript based on the suggestions given by him.

Reviewer #1 (Remarks to the Author):

This is a great manuscript, investigating the effect of expectation after pair learning on the encoding of information within a well-defined hierarchy of cortical areas in macaque monkeys, as measured with fMRI. The authors find that the encoding of the expected second images within pairs becomes more separable relative to the encoding of the first images within pairs. Furthermore, when the second image of a pair is unexpected, the encoding becomes even more separable. Furthermore, with an elegant analysis, they show that the features that make the encoding more separable indicates top-down processing within the hierarchy.

The design of study is creative and well thought out, the results are clearly presented, both in the figures and in the text, the methods are appropriately used, and the results are novel and highly relevant for the field.

We thank the reviewer for their positive remarks and furthermore, highlighting the elegance of our study design, analysis and findings of increased top-down pattern separation through predictions. We highly appreciate their helpful feedback on our interpretations of the neural mechanisms by which predictions aid distinguishing identities. We thank them for pointing out that predictions may exert their influence through complementary mechanisms like sharpening or dampening and agree with them on this point. We have incorporated this in our manuscript and provided a point-by-point response below. We hope with this we have addressed the important points raised by the reviewer.

I have only one point about the interpretation that I think could be a bit more nuanced. On lines 336-347, it is suggested that the results could also be interpreted in terms of dampening and sharpening. Some arguments against these interpretations are given, which are not all that convincing to me:

1) It is written: “Dampening predicts amplitude differences between expected and unexpected stimuli; while we find such differences in amplitudes in line with previous studies (Supplementary Fig. S2), they do not explain the differences in separability that we find across the face-processing hierarchy and between conditions, which arise from the multivariate spatial pattern of activity.”

However, the effect of the magnitude of the response for the different conditions (Fig. S2) seems nicely in line with the effect of separability (Fig. 2B), namely: higher for expected than ‘context-free’ and higher for unexpected than expected.

We thank the reviewer for this point and agree that the several proposed mechanisms like pattern separability, dampening, sharpening that underlie the benefits of predictions may indeed not be entirely mutually exclusive, but possibly complementary. The purpose of bringing up this point was to highlight that the increase in separability over the entire hierarchy and conditions is not a one-to-one mapping of magnitude differences. In fact, we observe higher amplitude for unexpected than expected in all the three face-areas (unexpected-expected: ML: $p=0.0003$; AL: $p=8.4570e-08$; AM: $p=1.8916e-06$), whereas only lower areas have higher separability in the unexpected than in the expected condition (ML: 77.1° vs. 61° , $p=2.631e-14$; AL: 82.8° vs. 77.2° , $p=8.084e-04$; AM: 89.6° vs. 87.5° , $p=0.1887$). Additionally, while lower areas in the hierarchy show higher separability in expected as compared to context-free (ML: 52.1° to 61° , $p=1.067e-12$; AL: 67.7° to 77.2° , $p=2.619e-07$), no difference in the amplitude is observed in these areas (expected-context-free: ML: $p=0.1747$; AL: $p=0.1457$). Our suggestion that pattern separability observed over the hierarchy and conditions is not (entirely) predicted by dampening - as assessed by response magnitude - is based on a Spearman correlation test between the pattern of results of average magnitude to that of average separability over the entire hierarchy in all conditions (Spearman’s $\rho=0.0167$, $p=0.9816$). This is done as such – while considering the entire hierarchy (i.e., all areas together) because pattern separation is inherently an inter-regional phenomenon¹ – an often ignored and under-appreciated property of pattern separation. Therefore, we consider the results over the entire hierarchy and not just in one of the face areas.

Further, we have now performed an additional analysis that has stemmed from the point raised by the reviewer in minor point 1) and are very grateful for this elegant idea. We found that magnitude *per stimulus* is not significantly correlated with the average separability of that stimulus with all the other stimuli in each condition and brain area separately (highest Spearman correlation’s $\rho = 0.10$; all $p>0.05$, corrected for multiple comparisons). Therefore, we find that pattern separability observed over the entire hierarchy and over all conditions is not explained by the magnitude of the population response through two analyses: 1) done over

the entire hierarchy and conditions, 2) done stimulus-wise for each condition and brain area separately. Nevertheless, we modify our interpretation and restrict our claim to magnitude differences observed and not to the mechanism of dampening. We instead state that dampening may be a complementary mechanism. The clarification has been provided in the results (line 203 – 206 in manuscript) and the discussion sections (line 379 – 391 in manuscript).

2) It is written: “sharpening accounts would suggest increased separability solely in AM because only those neurons are highly identity-selective.” There is some identity selectivity in ML and AL as well, as far as I know, which could then also be sharpened, right? In fact, it could be argued that sharpening could have a stronger effect in ML and AL than in AM, as the selectivity is farther from the noise ceiling?

We agree with the reviewer that some neurons in the face patches ML (19% of neurons) and AL (45% of neurons) are modulated by facial identity. This was shown to be significantly lower than the percent of neurons with identity modulation in AM (73% of neurons)². We are very thankful that the reviewer raises a very important point that indeed predictions could sharpen the identity selective neurons in ML and AL and agree that is certainly a possibility. Sharpening as a mechanism for expectation suppression suggests that expectation refines the inherent, existing representation of an area by suppressing the neural populations tuned away from the expected stimulus – effectively making the tuning narrower / sharper around the expected stimulus^{3–5}. Therefore, in the context of our study, we had inferred that if sharpening was at play, identity-specific predictions would result in increased separability in the regions with neurons possessing the highest identity selectivity, i.e., in AM. This would be analogous to results on sharpening of orientation representations in V1 but not V2 or V3³, because V1 contains neurons with a narrower tuning for orientation than higher visual areas⁶. But indeed, as the reviewer pointed out, the neurons in AM that are already highly identity-selective could hypothetically show a ceiling effect and hence little to no improvement of their representation. These are extremely important points that we had not considered and thank the reviewer for these. That said, we still think our results overall cannot be explained only by sharpening. An important tenet of ‘sharpening’ is that expectations refine *existing* representations³ by suppressing non-selective neurons. Our results depart away from this account of sharpening – because we observe a radical change in tuning properties exhibited. The radical switch from shape representation to mirror-symmetry observed in ML with expectations is not a mere enhancement of the existing (shape) representation in ML. Similarly, AL also shows appearance representation instead of its inherent mirror symmetric representation exhibited in the absence of predictions. We also found no signatures of mirror-symmetry in ML and similarly found no appearance representations in AL, when no predictions existed. Therefore, we agree with the reviewer that sharpening the identity-selective ML or AL neurons while suppressing non-selective neurons indeed could essentially increase separability as well. But it cannot explain the radical modification of representation towards high-level properties that we observe. Indeed, sharpening and top-down separability could be complementary mechanisms

and how these mechanisms interact with each other and when one or the other solution is preferred is an excellent question worth studying in the future. We have clarified this in the manuscript (line 379 – 391 in manuscript).

3) It is written that: “sharpening may be specific to early visual cortex and/or low dimensional stimuli such as gratings” This does not seem correct. In fact, initial work on sharpening of responses in monkey cortex showed the effects for natural objects in IT, see the work of Travis Meyer and Carl Olson, in particular: <http://www.nature.com/doi/10.1038/nn.3794>

Indeed! The suggestion that sharpening may be specific to early visual areas and/or low dimensional stimuli is not originally ours, but stems from previous human fMRI studies that found sharpening for oriented gratings only in V1 but not V2 or V3³ and no sharpening for visually presented objects in V1 or LOC⁷. This is in contrast to other studies, including the seminal work by Meyer and Olson (2011)⁸. We thank the reviewer for pointing this out. We have reformulated the section in question to clarify the origin of the argument, added a caveat to this interpretation and also cite the study in question (line 379 – 391 in manuscript).

In general, I would say that the finding of the increase separability is complementary to the ideas of dampening and sharpening. There does not seem to be any need to choose between one and the other as far as I can see. Furthermore, the current manuscript add on top of what is known, by showing that the effect of separability indicates top-down input, which could be true for dampening and sharpening as well.

In this respect, it could be relevant to add a reference to preliminary work by Koyano et al 2023 on dampening in different macaque monkey face patches: <https://www.abstractsonline.com/pp8/#!/10892/presentation/38672>

They find that familiarization only leads to long lasting changes in a high-level face patch (AM and not AF), which could lead to top-down modulation of these effects in lower-level face patches, if they would have been sensitive to these modulatory effects as in line the current study.

We thank the reviewer again for these very important interpretations of our results and for pointing us to this really interesting study on different timescales of plasticity across the face-processing hierarchy⁹ (which we have now also cited). The rapid and short-lasting plasticity observed in the lower area while slower timescale of the higher areas this is very much in line with the highly dynamic change in tuning and separability we observe in the lower areas of the hierarchy. We believe both these results indicate an intricate balance between stability vs flexibility in the network while internal models are refined with predictions or other learning processes. As clarified in the points above (line 379 – 391 in manuscript), we agree with the reviewer that indeed, amplitude decreases as a function of selectivity of neurons could be a complementary mechanism to top-down separability and believe it is an interesting avenue for future research.

Minor points :

- It is written: “we also found stronger responses in the unexpected than in the expected condition in all three face-areas (Supplementary Fig. S2; all $p < 3.5e-04$). However, this difference in magnitude did not explain the increase in pattern-separability (Spearman’s $\rho = 0.0167$, $p = 0.9816$) nor dimensionality ($\rho = 0.2667$, $p = 0.4933$)” It is not clear how this analysis is done. I assume that the correlation is calculated between the magnitude and the pattern-separability of the different stimuli, averaged across areas and conditions? Please clarify this in the text. Currently, it’s difficult to interpret this result.

As mentioned in major point 1) above – we had originally tested whether the mean amplitude over the stimuli and the mean separability across the hierarchy and conditions are correlated. We have now incorporated the suggestion from the reviewer and have performed an additional analysis per brain area and condition separately. Here, we tested whether higher magnitude in response to a given stimulus results in higher separability for that stimulus. We find that there is no significant correlation (highest Spearman correlation’s $\rho = 0.10$; all $p > 0.05$, corrected for multiple comparisons). With these two analyses, we infer that there is no one-to-one mapping between response magnitude and pattern separability. We thank the reviewer for suggesting such an elegant way of testing this relationship between pattern separability and magnitude differences. We have clarified the analysis steps in the manuscript as requested (line 203-206 in manuscript).

- It’s not clear how many times the different face identities are presented within a test block. If I understood correctly, there are 9 facial identity pairs and 12 additional unpaired facial identities? I assume that the unpaired faces are also presented during test blocks, during incongruent trials? If so, this might explain part of the effects, due to stimulus-specific adaptation, or dampening as the authors write. For all trials, the ‘context free’ condition are using one of only 9 faces. The amplitude of the response of these should then be lowest, as adaptation is strongest, and particularly for AM where identity selectivity is highest, which would be in line with the results in Fig. S2. Then, 60% of the trials are congruent, so the faces used for the expected condition appear more often than the faces used for the unexpected condition (assuming that there is a difference between the sets of faces used for these two conditions). This could again explain the effects shown in Fig. S2, where the effect of adaptation could be stronger for the expected condition than the unexpected condition. If this analysis is correct, it would be useful to add it to the discussion. In any case, it would be good to be more explicit about which facial identities are used for the different conditions, and how many times a facial identity is presented.

We have to clarify that there are nine trained face-pairs (shown in Supplementary S1) and there are no additional unpaired facial identities presented as violations. Identity violations of the predictions are made by presenting a different successor image from the same image set, i.e.,

a recombination of the successors from the trained pairs. Therefore, there is no difference in the overall exposure to the face identities between unexpected and expected conditions or any contribution of a novelty effect. The unexpected identity could only be chosen from the successors and a predictor image cannot be a successor. This ensures that predictions are only made forward in time and not as bi-directional associations between associations. The first image presented in a pair cannot be predicted from the previous image presented, hence it is free of predictive context. Each expected identity is presented 4 times in a test run. We have now expanded (line 546 – 547 in manuscript) on our previous description of the task design and hope that it helps to clarify the assignment of the face identities to the different conditions.

- I don't manage to find how much data was recorded. How many blocks per monkey? How many trials per block?

This information is contained in the methods section "Statistical Learning Paradigm". The testing phase of the statistical learning paradigm took place while fMRI data were acquired (Monkey L: 81 runs; Monkey P: 56 runs). In this phase, learned face-pairs were presented such that one face predicted the next; violations of the predictions (40% of all trials, total trials per run: 45) were introduced in the second image of a face-pair.

- It is not clear from the title, abstract nor figures that this is an fMRI study. I think it would be useful to make that explicit in the abstract, or in the first or second figure. I would also have expected to see some results at the whole brain level, at least the results for the localizer for both monkeys, to convince the reader that the different face patches could be reliably located in both subjects.

We thank the reviewer for pointing this out. We have now added this information to the abstract (line 4 in the manuscript). Also, the schematic of the study design in the figure 1 now explicitly indicates that this an fMRI study. Additionally, the whole-brain localizer results for both the monkeys have (figure below) have been added as a supplementary figure 1, as requested.

Figure R1: Face-areas localized in both monkeys using functional MRI. Significance maps from the contrast [faces vs. all other categories] are overlaid on coronal slices of anatomical MRIs. All localized face-areas are indicated with a white arrow (AM in the right hemisphere of Monkey P is not shown on this slice). Color bars indicate the negative log p-values of the significance map. Positive values of AP (in mm) indicate anterior from the ear-bars. (Supplementary Figure 1 in the manuscript)

Reviewer #2 (Remarks to the Author):

The authors study how contextual predictions influence the neural code in earlier visual regions during face recognition. Using fMRI in monkeys and focusing on the ventral visual system, the authors find that items that were expected or unexpected produced separable neural patterns in earlier areas that do not show such neural codes typically. This is an interesting study with clear results that advance our knowledge about predictive processing in the brain. I have some concerns about the clarity of the logic and some of the analyses.

We are happy to hear that the reviewer finds our study interesting and believes that our work advances our understanding of how predictive processing occurs in the brain. We are thankful to the reviewer for their careful assessment of our manuscript, raising important questions and providing constructive feedback. We believe their suggestions have made the manuscript much more thorough. Here, we have tried to address the points raised by the reviewer and hope that this helps provide clarity.

Major concerns:

1. I am not sure that the predictive items should be interpreted as baseline or context-free. Indeed, these items are not predicted. However, they do have predictive content – i.e., they predict the following items. This predictiveness has previously been shown to influence learning and memory as well as neural representations of items (e.g., Sherman et al., 2020, PNAS). In fact, one possibility is that predictive items, previously shown to be remembered at lower rates, might have low separability of their neural representations. Thus, it could be that rather than the predictability of items improving separation (in the Expected condition), it is the predictive cue (currently defined as baseline) that has lower than baseline separation. Another potentially preferable baseline might have been items that were interspersed between the pairs randomly, such that they are as familiar as other items, but are not predicted nor do not predict the following item. The question of an appropriate baseline is a difficult one, which might not have a straightforward answer, and it is impossible to change in the current experiment. I would recommend changing the conceptualization of the ‘baseline’ to a ‘predictive’ condition and modifying the interpretation of the results accordingly. To be clear, I still believe the results are important and interesting even under this interpretation.

We thank the reviewer for bringing up this important point. We fully agree that the question of an appropriate baseline is a difficult one and there are several possibilities, each with their pros and cons. The choice of the first image in a pair as a ‘context-free’ or baseline condition stems from the fact that it cannot be predicted from the preceding image. Furthermore, a previous behavioural study¹⁰ showed that visual sensitivity during the first image in a pair (within a structured stream) is the same as when the image is presented within a random sequence of images (when no predictability exists). Therefore, there is reason to believe that the first image in a pair within a structured stream is no different from an image embedded within a random sequence, which is usually considered “context free”. One of the possibilities for baseline could

have been to use the exact images prior to learning, presented in a random sequence – i.e. compare pre and post learning ^{10,11}. But due to time constraints and previous behavioural results from the lab indicating no difference between random images and first-image in a pair, we had chosen not to conduct a pre-learning fMRI. Another option for baseline, as suggested by the reviewer would have been to intersperse unpaired images already while training ¹². But since we had intended to ascertain learning of face-pairs through frequency-tagging of pupil dilation ¹⁰, we chose not to intersperse unpaired images with differential transitional probabilities during the training process. Therefore, to ascertain whether the first image in a pair is indeed ‘context-free’, we had compared the representations observed to ground-truth electrophysiology and active-appearance model. Our results (Fig. 3b, top panel) provide evidence that the predictor images have the same representation as predicted from electrophysiological recordings ^{2,13} in the respective areas when face images were presented to the monkeys without any context or learning effect.

The study ¹² pointed out by the reviewer shows a very interesting competitive dynamic between statistical learning effects - where predictions are based at the level of category - and subsequent episodic memory that is based on exemplars. The behavioural result of this study shows that predictive (1st image in a pair) scenes are remembered worse than the control unpaired scenes interspersed within a sequence of paired scenes and in one of the experiments also compared to the predicted. Indeed, as the reviewer suggests, this would indicate that statistical learning affects also the 1st image in a pair. However, we believe this result does not necessitate a change in ‘baseline’ of our study, because firstly the behavioural aspect of the study tests episodic memory (which relies upon hippocampal neural representations) and our study entirely focuses on the visual encoding representations in the IT cortex. Secondly, in their fMRI part of the study, they show that the behavioural difference between the 1st image and the control images is driven by the category decoding in hippocampus and change in decoding is exclusively present in hippocampus and not in visual areas. Significant category decoding is observed for the first, second image in the pair and the interspersed control images (Fig. S2) in visual areas (occipital cortex and category selective area, parahippocampal cortex). Although not statistically compared in that study, numerically it seems there is no clear difference in the decoding accuracy of the 1st image and the control image in both the visual areas. Similarly, there is no statistical main effect of condition (i.e., comparing the predictive and predicted). A subsequent study using intracranial recordings by the same authors furthermore shows that although some 1st images in a pair are subsequently forgotten, category information during the presentation of later forgotten and later not forgotten 1st images is identical, and hence, the authors conclude that variance in memory was not due to the strength of perceptual processing of the stimuli being encoded ¹⁴. It is the latter that we are determining in our context free condition. Therefore, while acknowledging that there are several ‘baseline’ possibilities, we also argue that in our study we do not need to change the ‘baseline’ and given the constraints of the current experimental design, the ‘context-free’ condition can be considered a valid baseline. Based on the reviewers’ suggestion, we have now incorporated this point in the discussion (line 392– 406 in manuscript).

2. Why was the condition of inducing prediction errors using different viewpoints included? It seems to reduce power and was not part of the main hypothesis. If that is for future analyses, or for analyses that did not pan out, or if this is a reanalysis of a previous dataset, this should be clearly laid out.

We included the unexpected view condition for several reasons: 1. We had used view violations in a previous electrophysiological study investigating prediction errors in face area ML¹⁵. In the manuscript at hand, we instead primarily focus on identities, which is a dimension closer to the computational goal of face processing and to which we did not have direct access in the previous study investigating only ML, which is primarily view/shape tuned. Nevertheless, we deemed the unexpected view condition a good comparison condition for the current fMRI results in the planning phase of this study. 2. While we focused primarily on identity in this manuscript, we did consider the unexpected view condition a good opportunity to try to internally replicate our results from the identity conditions. And indeed, we replicate one of our results regarding mirror-symmetric prediction errors from our previous electrophysiological study in ML; furthermore, we replicate a newly discovered decrease in mirror-symmetric tuning in AL that we also find in the expected condition as well as the unexpected identity condition. Since the unexpected view condition was part of the experimental design and the analyzed data, we consider it appropriate to report even if it is not the focus of the study, unless the reviewer suggests otherwise.

3. Please provide a clearer explanation or intuition of the math behind the dimensionality analysis. Specifically, it was hard to evaluate whether this measure is independent of the cosine distance, given that both cosine distance and eigenvalues are related to the covariance matrix of the data. I am concerned that this is not an independent analysis that warrants its own interpretation, rather than a different computation that captures similar aspects of the data like the cosine distance. This concern is reinforced by the high correlation between the measures.

We thank the reviewer for requesting clarification and further explanation on this point. We have now expanded the mathematical explanation along with the intuition behind dimensionality in the manuscript (line 617 – 619 in manuscript). We calculate the dimensionality of the neural space using participation ratio (PR): $(\sum_i \lambda_i)^2 / \sum_i \lambda_i^2$, where λ_i are the eigenvalues of the covariance matrix. This is done on the voxel \times stimuli trial-averaged matrix for each area and condition. High dimensionality (PR) indicates that the variance is spread evenly across the dimensions (see Fig. 2A of¹⁶ for a visual graphic)^{17–19}. To address the reviewer's question regarding independence of cosine distance and covariance matrix, we provide a proof-of-principle simulated example. For this, we generated a random matrix of the same size as the fMRI data and calculated its covariance matrix. We generated a second matrix whose covariance matrix is the same as the previous one's by adding a scalar value to the random matrix. Despite the two matrices having identical covariance, the cosine distances are different. Therefore, we clarify that cosine distance is not directly dependent on the covariance matrix (see Fig. R2 below). Nevertheless, we would like to clarify that dimensionality and

separability are indeed closely related concepts. Theoretical work ²⁰ shows that high-dimensional spaces facilitate separability of overlapping patterns by providing a larger activity space within which they are embedded in. This principle underlies the famous ‘kernel trick’ very commonly used in machine learning applications. Dimensionality is shown to indicate how the neural space is constrained better than pair-wise estimates of separability and takes into account the shape of the population-level distribution ^{18,21}. However, even within high-dimensional spaces, separability depends on the geometry of the variability of neural representations ²². Conversely, separability in a neural circuit can also be achieved through a sparse code ²³, expansion recoding ^{23,24}, or through non-linearly mixing of codes ²⁵.

With this, we hope that we have convinced and satisfied the reviewer that cosine distance and covariance matrix are not redundant measures. Dimensionality integrates several mechanisms that give rise to separability ²⁶ and provides information about the representational geometry that pair-wise measures do not provide and hence warrants its own interpretation.

Fig. R2: Different separability (cosine distance) despite same covariance matrices indicates the independence of these measures. Note different scaling in the left and right bar plot for visibility.

- 4. The logic of the PE-RDM analysis (reported starting in line 255) is unclear to me:
 - a. The properties captured by the RDM are the features of the faces (e.g., viewpoint, invariance etc.), not the PE signal which was computed as a difference between patterns. Why compare the difference between expected vs. unexpected, vs. just the representation of faces in the unexpected condition (as was done in the other analyses)?

We thank the reviewer for this question and the opportunity to clarify our approach. Our reply is distributed over the sub-questions a, b and c.

The primary reason to compute the difference between the expected and the unexpected conditions was to isolate the prediction error itself, and then to determine the content of this error signal. This is akin to the logic used in many studies on mismatch responses such as the mismatch negativity²⁷, which is defined as the difference between standards and oddballs. In fact, previous studies have used exactly the same logic as in our analyses, i.e., subtraction of expected and unexpected conditions, followed by multivariate analyses of the difference signal, to determine properties of prediction error responses²⁸. This approach rests upon the hypothesis that prediction errors can reflect more than general surprise, i.e., that they can carry information about the dimension in which the expectation violation has occurred. More on this in response to point b below.

b. It was unclear to me what this PE signal reflects. PE is usually thought of as a scalar value, potentially the difference in the average magnitude of the signal. How do the authors interpret this multivariate PE signal? Is that saying that each voxel has its PE signal, and looking at the pattern is a way to summarize that? Please clarify.

As stated above, we had set out to determine which information (if any) the prediction error signal might reflect. We start from the hypothesis that prediction errors on a population level signal content (and not merely surprise). In other words, prediction errors, especially in high-level cortex, are not just magnitude differences reflecting a surprise signal (indicating that an unexpected event has occurred). Rather, prediction errors can be considered as teaching signals reflecting content²⁹. Hence, prediction errors can signal *what* is unexpected rather than only that something unexpected has occurred. Feature-specific prediction errors have been found in several cortical and subcortical areas with neurons reflecting single feature deviance or a combination of different features^{15,30,31}. Furthermore, we found in a previous study that prediction errors are dependent on the feature-selectivity of the neurons emitting them¹⁵, which can be related to the concept of “precision” of prediction errors³². The premise behind our prediction error analysis is that individual neurons (or voxels in this case) may signal deviance based on their inherent tuning or specific feature axes, and hence, the population-level multivariate response reflects, e.g., a coherent learning signal. In fact, a recent study³³ showed that single neurons in the midbrain of rats express no content information when signalling a prediction error, but investigating prediction errors on a multivariate population level leads to a finding that content can indeed be decoded. Therefore, we wanted to understand how the content of PE is reflected at a population level in the brain regions we study.

Prediction errors are often studied as a relative change in activity of a neuron from the expected condition - here we aim to understand what this change in activity reflects on a population level. There are several accounts that prediction errors enhance the features of the sensory information³⁴, but on the other hand there are also accounts that prediction errors signal a correction from the internal model^{35–37}. By calculating change in voxel-level activity and then

investigating the population-level geometry, we had intended to find out whether PEs showed enhanced feedforward, i.e., local properties or higher-order properties (those of the predictions). We find that prediction errors are not just enhanced local information, but show higher-order signatures - similar to the prior. Furthermore, previous studies have shown that prediction errors are teaching signals with content – e.g., hippocampal PEs have been shown to both strengthen and update memories ²⁹. Therefore, we hope that we have convinced the reviewer that prediction errors are not simply a scalar value, but instead reflect rich content – coherently on a population level driving learning in the downstream region (line 270 – 279 in manuscript).

c. Mathematically, this analysis is not independent of the analysis in the Expected condition – because the Expected patterns are subtracted from the Unexpected. Thus, all the differences in similarity to RDMs might stem from the Expected patterns. Indeed, the patterns correlated. As currently done, I don't think this can be interpreted in a meaningful way separately from the Expected results.

Comparing unexpected conditions to expected conditions to identify prediction errors is common practice in several fields, e.g., in the literature regarding the mismatch negativity, but also in the literature investigating reward prediction errors in the dopaminergic system ³⁸. This stems from theoretical considerations that frame prediction errors as the result of a comparison between the expectation and the actual input. Hence, there is an intrinsic empirical and theoretical relationship between expected and unexpected conditions that is at the core of the hypothesized computation of prediction errors. We follow the same logic here. As stated above, combining the comparison of expected and unexpected conditions with a multivariate analysis is also not uncommon ²⁸. We should however clarify that when we reported a correlation between the expected and the unexpected conditions, we were referring to the correlation in the pattern of results of the 2nd level RSA across areas and tuning dimensions (i.e., the grey panel and the red panel in Figure 3B) – these patterns of results are correlated (albeit not perfectly), while the 1st level voxel patterns of the prediction errors and the 1st level voxel patterns of the expected condition are not (all $r < 0.30$, all $p > 0.07$). Given that the 2nd level RSA results are only imperfectly correlated and that the 1st level voxel patterns are uncorrelated, it is unlikely that the 2nd level RSA results in the unexpected condition are merely due to the way we identified prediction errors.

5. While I agree with the authors' general claim that properties of high-level regions are apparent in low-level regions in the Expected and Unexpected conditions, I would be more cautious with interpreting the results as meaning that these features 'cascade' or are transmitted from high-level regions. Currently, I believe that there is no data directly supporting this point in the paper. These features could emerge from altered internal properties, or they can be transmitted from earlier regions, or different regions not tested (e.g., the hippocampus or the prefrontal regions that are also important for predictive processing). To give some initial support for this part of their argument, I would recommend measuring functional connectivity between regions and testing if it is related to separability or the other neural effects reported.

For example, I imagine that there was some variance in the amount of separability between trials. The authors could leverage this variance and bin their trials (maybe to thirds) based on whether they are high/medium/low separability (in each experimental condition) in a lower area, and compute correlation in activity between areas (e.g., using beta-series connectivity, Rissman et al., 2004; or gPPI). The prediction is of higher connectivity in high vs. low separation. This wouldn't show directionality between regions, of course, or exclude the possibility that other regions are involved in orchestrating communication, but would provide initial support that communication between regions is related to separability.

We highly appreciate this point that the reviewer brought up and are thankful for their idea of assessing connectivity. Connectivity between the face-processing areas in macaque monkeys has been assessed in the past using several different methods like microstimulation³⁹, retrograde tracers⁴⁰, and resting state fMRI⁴¹. There are known feedforward and feedback connections between the face areas, but whether these areas show condition-dependent connectivity is a very interesting question. We have now conducted pattern connectivity analyses⁴² studying how much shared information exists between pairs of face areas. This measure of connectivity most directly relates to the analyses already reported in the manuscript. Based on our originally reported results, we hypothesize that pattern connectivity between the areas should be higher in expected condition than in the context-free condition, reflecting an increased interaction between areas in the former condition. For each condition, pattern connectivity was conducted as a repeated measures correlation⁴³ – where variance across the repeats, i.e., representations of each brain-area pair within the face-processing hierarchy was taken into account. Additionally, we controlled for spurious pattern connectivity by partialling out low-level similarities using the Gabor wavelet pyramid^{44,45}. Next, we tested whether there is an increase in pattern connectivity in the expected condition as compared to the context-free condition using a bootstrapped t-test. As hypothesized, we find that expected condition has a higher pattern connectivity between areas as compared to context-free condition (difference between correlations=0.2143, p=0.048). We conducted a similar test whether this holds true for unexpected condition as well and found an increase in pattern connectivity in the unexpected as compared to the context-free condition (difference between correlations=0.2036, p=0.047). Therefore, this increase in pattern connectivity between the face areas in expected and unexpected as compared to context-free condition strengthens our finding that indeed the high-level representations found in lower areas are a result of shared representations and connectivity between the areas. With this, we additionally provide evidence that connectivity between these face areas, which has been established for several years, can be modulated in a task-dependent manner. Together with our results from the RSA analyses, which are highly suggestive of top-down effects, the connectivity results may shed light on the functional role of the abundant feedback pathways found between the face areas⁴⁰. They also provide an interesting parallel to top-down effects on Kenyon cell odour responses in *Drosophila*, which are also thought to increase pattern separability⁴⁶. We have now incorporated these results in the manuscript (line 264-267; 330-331; 712-720 in manuscript)

and thank the reviewer again for helping us make our manuscript better through the peer-review process.

Minor concerns:

1. Line 27: I think the sentence is nongrammatical. Should be “The reason for why we struggle...” or “The reason for struggling to distinguish...”

Thanks for noting. Corrected to “The reason for why we struggle” (line 27 in manuscript).

2. Line 35: the hypothesis that “incorporating higher-order predictive information leads to more separable (Fig. 1A) and invariant (Fig. 1B right) representations already in lower areas of the hierarchy” Is not substantiated. I understand everything that precedes that hypothesis, but it does not lead to the hypothesis. Why would that be the solution? For example, it can be that face recognition is done in such cases based on higher-order areas that have separable neural patterns. Or why would predictions from higher-order areas that are communicated to lower-order areas lead to separable codes? Is it because there are separable neural populations activated in these higher-order areas, thus they communicate, and activate separable populations in lower-order areas? As currently stated, the logic is unclear.

We thank the reviewer for spotting this omission. Predictive processing theories suggest that predictions are generated in higher areas and take effect in lower areas in processing hierarchies. Predictions should hence reflect the tuning properties of the areas that generate the predictions, i.e., of higher-level areas. We hypothesize that if predictions are transmitted from higher to lower areas, they carry this computational format, and hence also the inherently higher separability and invariance properties of higher areas with them. We have now clarified this in the introduction (line 35 – 38 in manuscript).

3. Given that the authors use a non-invasive fMRI method to study their question, I think it is worth mentioning why choosing a model animal like monkeys, rather than directly testing the question in humans (which are easier to test and have a well-characterized face processing system). Is it an intermediate step in translating electrophysiology research in monkeys to human fMRI (and if so, were the findings reported in the paper previously reported in electrophysiology)? Or another reason? This might be self-evident for monkey researchers, but less so for the broad audience of Nature Communications.

Certainly an important point! We have now included this in the manuscript (both in the introduction (line 50-51 in manuscript) and methods section (line 448 – 457 in manuscript)) based on the input from the reviewer. We believe that both predictions⁴⁷ and pattern separation¹ for identity recognition are network-level phenomena involving interactions between hierarchical areas such that a representational transformation occurs between the input and outputs of the areas. Investigating the neural mechanisms underlying these processes requires studying face processing on the network level, which fMRI gives access to. Conducting such a study while having simultaneous access to all three areas using

electrophysiology is technically challenging and even impossible in some animals because of the complex blood vessel pattern in the STS that may preclude simultaneous recordings. On the other hand, studying representational change within the face-processing hierarchy in non-human primates using fMRI has distinct advantages: 1. It allows direct comparison to the representation known from ground-truth electrophysiology data ^{2,13}. 2. Anatomical connectivity about direct feedforward and feedback connections ⁴⁰ between these face areas along with functional connectivity ⁴¹ has been mapped out; 3. A clear hierarchical organization between these areas has been determined ^{2,39}. Such ground-truth data on connectivity, representations, and hierarchy are not available in humans: even though a face-processing network comprising areas in the occipital lobe, fusiform gyrus, and anterior temporal lobe have been found ⁴⁸, how these areas' representations differ or overlap remains a matter of debate ⁴⁹⁻⁵², as much as their hierarchical arrangement ⁵³⁻⁵⁵. Having ground-truth information about the face-processing representations in NHPs available and conducting this study with fMRI has aided us in having extremely precise hypotheses about representational changes and made interpretation of our results very clear. This does not preclude future electrophysiological studies or studies in humans, but provides an important puzzle piece in unravelling predictive processing in higher-order visual cortex.

4. Figure 1B: What are S1, S2, S3? Why not use ML, AL, AM?

S1,2,3 refers to stage 1, 2, and 3 in a processing hierarchy. The intention was to signify that the hypothesis about hierarchical prediction passing and the resulting changes in separability may apply more generally to hierarchically organised systems outside of the face-processing areas as well. The figure legend now states what S1, 2, 3 signify. If the reviewer prefers, we can also change the figure and caption to ML, AL, AM.

5. Line 118: I'm not sure I agree that "This suggests that high separability otherwise characteristic of the top of the processing hierarchy can already be attained in lower areas if predictive information exists - potentially facilitating readout of information from well-decorrelated representations." The authors show an increase in separability, but regions ML and AL are still about 10-20 degrees less separable than AM. Why do the authors argue that they achieve high separability like at the top of the processing hierarchy?

We fully agree with the reviewer that the average separability in ML and AL does not reach the same level as that observed in AM (although some identities become orthogonal or almost orthogonal to each other even in ML). We did not mean to imply with this statement that the separability is identical. We have therefore reformulated the sentence in question: "This suggests that high separability otherwise characteristic of the top of the processing hierarchy can be approached in lower areas if predictive information exists - potentially facilitating readout of information from well-decorrelated representations." We have also changed a similar formulation in the abstract (line 8, 124 in manuscript).

6. Line 122: the logic of why unexpected faces would increase pattern separation more than expected faces is unclear to me. The authors write: "Predictive processing theories suggest that

predictions serve to compute prediction errors (PE) as a deviation between the prediction and the sensory input to reduce redundancy in information transmission: redundant information is removed, optimizing the dynamic range of neurons and allowing them to efficiently signal decorrelated or unpredicted information. In terms of pattern separation, this may entail higher pattern-separability when PEs (to unexpected faces) occur.” I agree that PE should reduce redundancy in information, and that could lead to separate representation. However, that should promote learning and predictability, such that the expected faces should be those for which the redundancy/noise has already been removed, and separability is highest. Please explain. I’m not arguing that there couldn’t be reasons why PE would increase separability, and potentially more than expected stimuli (e.g., Frank et al., JNeuro, 2020; Kim et al., JNeuro, 2017; Bein et al., 2023), but that I don’t understand the logic of the argument laid here.

We thank the reviewer for giving us the opportunity to clarify the reasoning behind our hypothesis. Our hypothesis regarding pattern separability in the unexpected condition is driven by theoretical considerations from the predictive processing literature that suggest that prediction errors signal differences between expected and unexpected events. Specifically, it has been suggested that neurons compute the difference between the predicted activity level and the actual activity – this allows utilizing the entire dynamic range of the neuron to encode only a small range of signal levels. This sparse coding regime allows signalling a larger number of distinguishable images than other coding regimes⁵⁶. When we consider ML and AL, we find an increase in pattern separability in the unexpected condition that shows a statistically significant increase over the expected condition; this suggests that the highest separability in face processing areas in visual cortex is not necessarily achieved in the expected condition, at least not in our paradigm (but higher decoding for unexpected than expected stimuli has for example also been observed in IT cortex⁵⁷ and using EEG in humans⁵⁸, as much as in the literature on memory updating, where it has been suggested that prediction violations may lead to the creation of new memories in specific hippocampal subfields, separated from existing memories - although integration may also take place; see⁵⁹ for an encompassing review). We have previously reported that the largest prediction errors in ML are elicited by neurons which are highly sparse and very narrowly tuned¹⁵. Translated to the population level in our data at hand, this would suggest that the coding regime utilized in the unexpected condition allows for an optimization of the available dynamic range of the neurons in AL and ML to signal differences from the predicted face, which are more separable than, e.g., the images without predictive context or in the expected condition where no prediction errors are computed. Alternatively or additionally, it is possible that neurons emitting prediction errors in the unexpected condition add additional feature dimensions to the neural population activity, e.g., through non-linear mixed selectivity for surprise and stimulus features. This would increase the dimensionality of the population response, which would, in turn, increase the separability of stimuli²⁵. Hence, there are multiple, not mutually exclusive theoretical possibilities why the unexpected condition can show a higher separability than the expected condition in our data. We have reformulated the section in question as well as part of the discussion to make this point clearer (line 131-132; 367-371 in manuscript).

7. Line 226: I'm not sure why this is the first introduction to the analysis as an RSA analysis. The cosine distance analysis performed earlier in the paper is also a type of RSA. Please consider moving that definition and ref earlier.

Following the reviewer's suggestion, we moved the definition and citation from line 226 to line 101, i.e., when introducing the analysis on pattern separability using cosine distances.

8. Figure 3, Aii is confusing, because it seems like a step in the analysis between Ai and Aiii, whereas, if I understood correctly, it is not (for the context-free and expected conditions). I'd recommend moving it to next to the Prediction error panel in B (or changing the analysis altogether, see major comment 4).

Panel Aii in Figure 3 indeed illustrates the analysis of the unexpected condition. Panel Aiii illustrates the 2nd level representational similarity analyses, which is performed both in the expected and the unexpected condition. We opted not to move Panel Aii next to the prediction error panel in B because this would separate the computation of prediction errors from their subsequent analyses by means of RSA. It would also create a lot of empty space in the figure that would make the figure elements very small and thus hard to read. To accommodate the reviewer's comment, we clarified in the figure caption of panel A (line 214-215 in manuscript) that 2nd level RSA was applied to the expected and the unexpected condition, respectively.

9. In the Results, section "high-level predictions cascade down the face-processing hierarchy", please describe briefly what is the known electrophysiological tuning property of each region (based on the figure and later in the text, I believe it is view-specificity/mirror symmetry/view invariance). It would be useful to include it already in the initial description of the electrophysiological tuning. Additionally, please describe shortly how would that translate to an RDM – e.g., if view invariance, was the RDM capturing high similarity between different viewpoints of the same face, but lower similarity between different faces?

We thank the reviewer for bringing up this point and have now refined it in the manuscript. We have to clarify that the RDMs used were based on recent papers^{13,60} which show that face area ML represents face shape, while AM at the top of the face processing hierarchy represents facial appearance. Previous studies found that higher hierarchical areas have higher view-invariance than lower areas in the face-processing network². We computed these measures based on the 9 different identities in overall (not each) 3 different views. Since the faces-pairs were trained only with specific images (i.e., the same view), we could not make a model RDM with all the different views of each identity. Hence, the appearance model contains a component of the facial view. Therefore, we additionally computed a model RDM using only the frontal-view versions of the same nine facial identities and performed the same procedure to obtain the hypothetical (view-independent) appearance RDM. Although this hypothetical RDM is made with only frontal faces, correlating it with an empirical RDM based on three different views is what makes this a view-independent/invariant appearance model (line 341 in manuscript).

10. Line 249: why would predictability reduce the view-independent appearance in AM? How do the authors explain that?

We provide an interpretation of the reduction in view-independent appearance tuning in lines 340-343 in the discussion section of the manuscript. In brief, if we extrapolate the results from the three-level hierarchy of ML, AL, and AM that we investigate here, it is conceivable that the same principles that we found at these three stages also hold for subsequent processing stages. Specifically, we would expect areas on a hierarchically higher level than AM, possibly outside the face processing system, to pass down tuning properties to AM. In keeping with what we found in AL and ML, this would also entail a reduction in the feedforward tuning properties of AM, i.e., appearance tuning. Ultimately, this systematic shift of tuning properties would free resources at the very top of the processing chain to perform computations that would otherwise not be possible during feedforward processing. To clarify the link between the results reported in line 249 and the discussion in lines 340-343, we now explicitly mention the reduction in view-independent appearance tuning in AM in the latter section.

11. Line 536: was the multiple correction done for 3 ROIs? Or the number of comparisons?

The remark about multiple comparisons correction in line 536 is a general statement about the method that we used to correct for multiple comparisons throughout the manuscript⁶¹, not specifically for analyses that need to be corrected across regions of interest. For example, when comparing pattern separability for identities between conditions in lines 114ff, we correct for the number of regions of interest, because this corresponds to the number of tests reported. When we compare participation ratio to the noise ceiling per region of interest and conditions in lines 174ff, the p-values are corrected for 9 comparisons (regions × conditions) using the same method.

12. Please clearly state in the Methods what was the sample for all tests. For example, in line 116, what does 70 df's reflect? Or in line 136, what do 34 df's reflect?

We have added the requested information to the methods section and elsewhere in the manuscript (highlighted). Regarding the specific requests of the reviewer, both tests compare between stimuli. E.g., the 70 degrees of freedom in line 70 reflect comparisons between stimuli (unpaired test, $df(2,70) - \text{number of comparisons} = 36$, total $36 \times 2 = 72$).

Reviewer #3 (Remarks to the Author):

This manuscript investigates how contextual information affects the neural representation of faces in three face areas of the macaque brain using functional MRI. After exposing the animals to temporal associations of nine pairs of faces, they found that the representation of faces depends both on the temporal order of the presentation and on whether the successor face can be predicted by the associations. Their results suggest that the responses to successor faces reflect “top-down” signals from higher-level face areas. Overall, this is a well-conducted study, with new insights into how statistical learning affects neural representation. Although there’s a lot to like in their paper, I have a hard time understanding the computational implications of the results. My specific comments are as follows:

We thank the reviewer for assessing our work as a well-conducted study and are happy to hear that the reviewer found our work on how top-down predictions acquired through statistical learning modify neural representations insightful. In the revised version of the manuscript, we have expanded on the computational implications of having invariant, high-level predictions accessible early on the hierarchy as well as the benefits of dynamic context-dependent switching of population tuning properties through predictions. We hope that these additions, as well as the requested control analyses (see below) have improved the manuscript to the reviewer’s satisfaction.

Major:

1) The interpretation of the results. The senior author of this paper previously used this paradigm in macaques and claimed that lower-level face regions compute prediction errors based on prediction signals from higher-level face areas (Schwiedrzik and Freiwald, 2017). In this paper, the comparison between expected and unexpected stimuli can also be explained by the predictive coding framework. However, it’s unclear to me how the comparison between expected and context-free stimuli fits into the picture. There should be no error for the expected stimuli, so what’s the nature of the representation? The current narrative focuses on explaining the what the data look like, but a single coherent framework is not in sight.

The reviewer is absolutely correct in pointing out that we investigated prediction errors, a core computation in predictive processing theories, in this previous study¹⁵. In the manuscript at hand, we focus primarily on the mechanisms and consequences of predictions themselves. We develop an interesting link between context, top-down processes, and pattern separation, that would not be predicted by classical predictive processing theories. Hence, we see this as a first step in building a new framework that takes a different angle at the question of predictive processing. Without doubt, the question of the nature of the representation in the contextually predicted condition that the reviewer points out is at the heart of this, as well as the differences to the unexpected condition where prediction errors are computed. Prediction errors are a core component of predictive processing theories and hence the unexpected condition is

important in assessing the new hypotheses we put forward in light of this computation. Regarding the nature of the representation in the contextually predicted condition, we hypothesize that predictions are passed from higher to lower areas and carry with them the representational format of the area where the predictions are generated. This is in principle compatible with predictive processing theories (that however usually remain mute on the question of the specific representational format of predictive messages). Our data shown in Figure 3 support our hypothesis. We have also added a pattern connectivity analysis between the face areas to the manuscript that reveals that indeed, there is an increase in connectivity in the expected condition as compared to the context-free condition (see lines 264ff, difference between correlations=0.2143, $p=0.048$). This further strengthens our hypothesis that the high-level representations found in lower areas result from functional interactions between the hierarchical levels of the face processing system. In the unexpected condition, we would expect the same top-down messages to be at play, but in addition, prediction errors should be computed. To analyze the unexpected condition, we isolate prediction errors by first subtracting the unexpected from the expected condition, and then analyze the representational format of the prediction errors. We indeed find a similar (but not identical) tuning profile as in the expected condition, which is in line with our previous electrophysiological findings on prediction errors only in ML, which also showed representational signatures of the predictions from which they were computed (e.g., mirror-symmetry¹⁵). We also find higher pattern connectivity between areas in the unexpected as compared to the context-free condition (difference between correlations=0.2036, $p=0.047$), in line with hierarchical message passing being at the core of the computation of prediction errors. What we critically add with our contribution and what goes beyond our and others' previous work is the relationship of top-down predictive message passing and pattern separability in cortex. We find that top-down predictions increase pattern separability in lower face processing areas, assigning a new functional role to predictions. Furthermore, pattern separability is even higher in the unexpected than in the expected condition, which we interpret to possibly be a consequence of a different coding regime in the unexpected condition that involves contrastive coding and dynamic range optimization as in classical predictive coding theories⁵⁶, and/or non-linear mixed selectivity²⁵ between surprise and feature-specific signals, which could explain the increases in dimensionality that we found (Figure 2c). We thus hope to be on a good way towards an encompassing framework that combines theoretical concepts from predictive processing and pattern separation.

2) A surprising result in Figure 3 is that while lower-level face areas inherit representation from higher-level areas, they also discard their original representation. I'm puzzled by this result because one would expect the higher-level representation to come from a feedback signal, but that representation would have to be in the higher-level area first in order to be transferred to the low-level area via feedback later on. This conflict may be resolved if the feedback signal comes only from the context-free predictor, but the feedforward signal should still be present so that the prediction error can be computed. Relatedly, in the last row of Figure 3, which faces are used to compute the similarity matrix, the successor faces in the unexpected condition? If

so, the representation cannot come from the predictor face, and I would expect to observe the representation at the original level. Alternatively, the loss of the original representation format may be due to the lack of temporal resolution of the fMRI technique. In any case, this should be thoroughly discussed.

We fully agree with the reviewer that the predictor face likely activates top-down processes that then act upon the successor face via feedback signals. The time delays between our stimuli render this scenario absolutely plausible, i.e., there is sufficient time for the predictor face to activate predictions in higher areas that are then tested against the incoming successor in lower areas. As the reviewer points out, we find signatures of prediction errors in the unexpected condition, which are computed in the representational format of the predictions, i.e., of the higher area from which they originate, in line with this interpretation. To clarify the analysis of the unexpected condition, we point out that we test in which format prediction errors to successor faces are represented using 2nd level RSA between the prediction errors to successor faces and a model (and not between the successor and predictor). We fully agree with the reviewer that fMRI neither has the spatial nor temporal resolution to isolate all the computational components that are likely at work within a population of neurons that is thought to be minimally composed of prediction error and prediction units which are distributed, e.g., over different layers⁶². Our previous electrophysiological results¹⁵, which were however restricted to a single area ML, also suggest at least two temporal phases of information passing: an early phase with a latency of about 130 ms in which prediction errors are computed which are dominated by local tuning properties; and a later phase where recurrent processing across hierarchical levels may take place and where high-level tuning properties like mirror-symmetry become apparent in ML prediction errors. Hence, further electrophysiological studies are needed to dissect the subcomponents, subpopulations, and time dynamics of predictive computations in the face processing system. However, we provide a proof of principle about the underlying computations on the population level, which replicates across the entire face processing hierarchy, and which we hope will inspire these kinds of studies in the future. We have added this discussion to the manuscript (line 347 – 361 in manuscript) as suggested and thank the reviewer for raising this important point.

3) What is the composition of the nine face pairs? Will the same face appear as both a predictor and a successor? If not, then the face pairs being compared are not perfectly matched. This may affect the results in Figure 2. For example, the nine successor faces may be more separable from each other in the neural space than the nine predictor faces. Please clarify the procedure and estimate the size of the effect.

The composition of the nine face pairs has been clarified in the modified supplementary figure 2a (see below). The nine pairs are composed of 18 unique identities, where both the predictors and successors are counter-balanced for different views and gender. The same face does not appear as both a predictor and successor. This was done to elicit only predictions forward in time not confounded with bi-directional or symbolic associations⁶³. Indeed, as pointed out by the reviewer, the comparison of context free and expected/unexpected is between two

different sets of faces – hence all statistical tests done for Figure 2 between predictor and expected/unexpected successor are unpaired tests. We note that the images were all luminance equated to eliminate potential low-level differences. We have now performed additional controls on the image material that show that there are no separability differences between the set of predictor and successor face-stimuli in terms of low-level image properties (gabor-filterbank, $p = 0.7255$; t-stat: -0.3525 ; df: 70) as much as in terms of shape ($p = 0.5494$; t-stat: 0.6015 ; df: 70) and appearance ($p = 0.3682$; t-stat: -0.9056 ; df: 70). Therefore, stimulus-level properties do not explain the increased neural separability in the expected/unexpected condition over the context free condition that we observed in the face areas. We thank the reviewer for this question and we have now added this important piece of information to the manuscript (line 511 – 514 in manuscript).

Fig. R3 (also Supplementary Figure 2A): Trained face-pairs counter-balanced for the different views. The face images were created using FaceGen Modeller

Minor:

1) Last line on page 4: “Pattern-separability increased along the hierarchy (ML<AL<AM)”. Where is the data shown, Figure 2B? Please refer to the figure.

Thanks for noting. Indeed, we were referring to Figure 2b. This has now been corrected.

2) In several cases, p-values of statistical tests were reported in the figure legends and in the main text, but detailed information, including the type of statistical test and the degree of freedom, was missing. Please provide this information.

We have now made sure that the type of test is mentioned throughout the results section (see highlighted), as well as degrees of freedom (where appropriate).

3) Page 7, line 187. The correlation between increases in dimensionality and separability across conditions and hierarchy was computed and tested for statistical significance. Since each value here is a difference between two values that could appear in other comparisons, the samples cannot be considered independent. Is the statistical test corrected for this?

We thank the reviewer for raising this point because it alerted us to an imprecise formulation in reporting this result, which we now corrected (line 194 in manuscript). While we stated that the increases in dimensionality and separability were correlated, what we actually meant was that dimensionality and separability were correlated across areas and conditions. We apologize for this mistake. Given this, the problem that the reviewer points out does not apply, since there is no overlap between the data entering the correlation analyses stemming from comparisons between conditions, and the test statistic we report is appropriate for paired data. Again, we apologize for this mistake.

4) As pointed out by Dubois et al. 2015, information that can be decoded from single-unit responses may not be available in fMRI response patterns. This technical limitation should be discussed by the authors.

We thank the reviewer for this point. Indeed, we replicate the known electrophysiological properties of AM, i.e., appearance tuning, using fMRI response patterns. As the reviewer correctly points out, Dubois et al. (2015)⁶⁴ found a discrepancy between single neuron tuning and multivariate pattern analyses in AM (while their results were convergent in other face areas, especially ML). Specifically, these authors were unable to decode identity information from AM fMRI data, when this information is readily apparent on the single neuron level in the same brain area. Dubois et al. suggest that decodability from fMRI data may depend on the particular spatial organization of the variable being decoded. Yet, spatial clustering for head orientation, which was decodable in AM, did not differ from spatial clustering for identity, which was not decodable in that study. Hence, it is not clear that there is a fundamental limitation grounded in physiological properties of AM that prevents retrieving electrophysiological ground truth information. However, there are several technical differences between the study by Dubois et al.⁶⁴ and our study that may explain why these authors did not replicate electrophysiological tuning properties using fMRI while we did: 1. We used an event-related design (images were presented for a 0.5 s with long baselines in between and were temporally jittered such that response patterns could be extracted most optimally), while Dubois et al. used a block design. Temporal blurring and/or signal saturation in a block design may prevent distinguishing responses to different stimuli. 2. We assess appearance tuning using 2nd level representational similarity analyses, while Dubois et al. decoded discrete

identities in a binary (one-to-all fashion) using MVPA. Hence, both the dimension as well as the precise method differ between studies. 3. We acquired 81 and 56 runs per animal, while Dubois et al. acquired on average 17 (range 10 to 38) runs per animal. 4. There are several other unmatched or potentially unmatched parameters such as the scanner being used (TIM Trio in Dubois et al., Prisma FIT in our study), contrast agent (MION used in Dubois et al., not used for the main experiment in our study), coil (no information given in Dubois et al.), coil positioning (no information given in Dubois et al.), etc. Given these multiple differences, it is not possible to precisely pinpoint which of these parameters may have contributed to the discrepancy between studies. Given the overall difficulty in interpreting negative results, we respectfully wish to refrain from discussing the (technical) merits of the work by Dubois et al. in our manuscript (unless the reviewer deems it absolutely necessary). We point out that Dubois et al. concluded “Further studies are needed to elucidate the relationship between information decodable from fMRI multivoxel patterns versus single-unit populations”.

References:

1. Santoro, A. Reassessing pattern separation in the dentate gyrus. *Front. Behav. Neurosci.* **7**, (2013).
2. Freiwald, W. A. & Tsao, D. Y. Functional Compartmentalization and Viewpoint Generalization Within the Macaque Face-Processing System. *Science* **330**, 845–851 (2010).
3. Kok, P., Jehee, J. F. M. & de Lange, F. P. Less Is More: Expectation Sharpens Representations in the Primary Visual Cortex. *Neuron* **75**, 265–270 (2012).
4. de Lange, F. P., Heilbron, M. & Kok, P. How Do Expectations Shape Perception? *Trends Cogn. Sci.* **22**, 764–779 (2018).
5. Yon, D., Gilbert, S. J., de Lange, F. P. & Press, C. Action sharpens sensory representations of expected outcomes. *Nat. Commun.* **9**, 4288 (2018).
6. Vogels, R. & Orban, G. A. Activity of inferior temporal neurons during orientation discrimination with successively presented gratings. *J. Neurophysiol.* **71**, 1428–1451 (1994).
7. Richter, D., Ekman, M. & de Lange, F. P. Suppressed Sensory Response to Predictable Object Stimuli throughout the Ventral Visual Stream. *J. Neurosci.* **38**, 7452 (2018).
8. Meyer, T. & Olson, C. R. Statistical learning of visual transitions in monkey inferotemporal cortex. *Proc. Natl. Acad. Sci.* **108**, 19401–19406 (2011).
9. Koyano, K. W. *et al.* Distinct temporal scales of plasticity in macaque AM and AF face patches. in *The Functional Organization and Plasticity of the Ventral Stream of the Visual System* (2023).
10. Schwiedrzik, C. M. & Sudmann, S. S. Pupil diameter tracks statistical structure in the environment to increase visual sensitivity. *J. Neurosci.* JN-RM-0216-20 (2020) doi:10.1523/JNEUROSCI.0216-20.2020.
11. Schapiro, A. C., Kustner, L. V. & Turk-Browne, N. B. Shaping of Object Representations in the Human Medial Temporal Lobe Based on Temporal Regularities. *Curr. Biol.* **22**, 1622–1627 (2012).
12. Sherman, B. E. & Turk-Browne, N. B. Statistical prediction of the future impairs episodic encoding of the present. *Proc. Natl. Acad. Sci.* **117**, 22760–22770 (2020).
13. Chang, L. & Tsao, D. Y. The Code for Facial Identity in the Primate Brain. *Cell* **169**, 1013-1028.e14 (2017).
14. Sherman, B. E. *et al.* Temporal Dynamics of Competition between Statistical Learning and Episodic Memory in Intracranial Recordings of Human Visual Cortex. *J. Neurosci.* **42**, 9053 (2022).
15. Schwiedrzik, C. M. & Freiwald, W. A. High-Level Prediction Signals in a Low-Level Area of the Macaque Face-Processing Hierarchy. *Neuron* **96**, 89-97.e4 (2017).
16. Litwin-Kumar, A., Harris, K. D., Axel, R., Sompolinsky, H. & Abbott, L. F. Optimal Degrees of Synaptic Connectivity. *Neuron* **93**, 1153-1164.e7 (2017).
17. Peiran Gao *et al.* A theory of multineuronal dimensionality, dynamics and measurement. *bioRxiv* 214262 (2017) doi:10.1101/214262.
18. Recanatesi, S., Ocker, G. K., Buice, M. A. & Shea-Brown, E. Dimensionality in recurrent spiking networks: Global trends in activity and local origins in connectivity. *PLOS Comput. Biol.* **15**, e1006446 (2019).

19. Recanatesi, S., Bradde, S., Balasubramanian, V., Steinmetz, N. A. & Shea-Brown, E. A scale-dependent measure of system dimensionality. *Patterns* **3**, 100555 (2022).
20. T. M. Cover. Geometrical and Statistical Properties of Systems of Linear Inequalities with Applications in Pattern Recognition. *IEEE Trans. Electron. Comput.* **EC-14**, 326–334 (1965).
21. Cayco-Gajic, N. A., Clopath, C. & Silver, R. A. Sparse synaptic connectivity is required for decorrelation and pattern separation in feedforward networks. *Nat. Commun.* **8**, 1116 (2017).
22. Chung, S., Lee, D. D. & Sompolinsky, H. Classification and Geometry of General Perceptual Manifolds. *Phys. Rev. X* **8**, 031003 (2018).
23. Marr, D. A theory of cerebellar cortex. *J. Physiol.* **202**, 437–470 (1969).
24. Albus, J. S. A theory of cerebellar function. *Math. Biosci.* **10**, 25–61 (1971).
25. Rigotti, M. *et al.* The importance of mixed selectivity in complex cognitive tasks. *Nature* **497**, 585 (2013).
26. Cayco-Gajic, N. A. & Silver, R. A. Re-evaluating Circuit Mechanisms Underlying Pattern Separation. *Neuron* **101**, 584–602 (2019).
27. Garrido, M. I., Kilner, J. M., Stephan, K. E. & Friston, K. J. The mismatch negativity: A review of underlying mechanisms. *Clin. Neurophysiol.* **120**, 453–463 (2009).
28. An, H., Ho Kei, S., Auzztulewicz, R. & Schnupp, J. W. H. Do Auditory Mismatch Responses Differ Between Acoustic Features? *Front. Hum. Neurosci.* **15**, (2021).
29. Sinclair, A. H., Manalili, G. M., Brunec, I. K., Adcock, R. A. & Barense, M. D. Prediction errors disrupt hippocampal representations and update episodic memories. *Proc. Natl. Acad. Sci.* **118**, e2117625118 (2021).
30. Audette, N. J. & Schneider, D. M. Stimulus-Specific Prediction Error Neurons in Mouse Auditory Cortex. *J. Neurosci.* **43**, 7119 (2023).
31. Oemisch, M. *et al.* Feature-specific prediction errors and surprise across macaque fronto-striatal circuits. *Nat. Commun.* **10**, 176 (2019).
32. Friston, K. A theory of cortical responses. *Philos. Trans. R. Soc. B Biol. Sci.* **360**, 815–836 (2005).
33. Stalnaker, T. A. *et al.* Dopamine neuron ensembles signal the content of sensory prediction errors. *eLife* **8**, e49315 (2019).
34. Tang, M. F. *et al.* Expectation violations enhance neuronal encoding of sensory information in mouse primary visual cortex. *Nat. Commun.* **14**, 1196 (2023).
35. Revina, Y., Petro, L. S. & Muckli, L. Cortical feedback signals generalise across different spatial frequencies of feedforward inputs. *New Adv. Encoding Decod. Brain Signals* **180**, 280–290 (2018).
36. Papale, P. *et al.* The representation of occluded image regions in area V1 of monkeys and humans. *Curr. Biol.* **33**, 3865–3871.e3 (2023).
37. Bi, Z., Li, H. & Tian, L. Top-down generation of low-resolution representations improves visual perception and imagination. *Neural Netw.* **171**, 440–456 (2024).
38. Eshel, N. *et al.* Arithmetic and local circuitry underlying dopamine prediction errors. *Nature* **525**, 243–246 (2015).

39. Moeller, S., Freiwald, W. A. & Tsao, D. Y. Patches with Links: A Unified System for Processing Faces in the Macaque Temporal Lobe. *Science* **320**, 1355–1359 (2008).
40. Grimaldi, P., Saleem, K. S. & Tsao, D. Anatomical Connections of the Functionally Defined “Face Patches” in the Macaque Monkey. *Neuron* **90**, 1325–1342 (2016).
41. Schwiedrzik, C. M., Zarco, W., Everling, S. & Freiwald, W. A. Face Patch Resting State Networks Link Face Processing to Social Cognition. *PLOS Biol.* **13**, e1002245 (2015).
42. Kriegeskorte, N., Mur, M. & Bandettini, P. Representational similarity analysis - connecting the branches of systems neuroscience. *Front. Syst. Neurosci.* **2**, 4 (2008).
43. Bakdash, J. Z. & Marusich, L. R. Repeated Measures Correlation. *Front. Psychol.* **8**, (2017).
44. Tai Sing Lee. Image representation using 2D Gabor wavelets. *IEEE Trans. Pattern Anal. Mach. Intell.* **18**, 959–971 (1996).
45. Kay, K. N., Naselaris, T., Prenger, R. J. & Gallant, J. L. Identifying natural images from human brain activity. *Nature* **452**, 352–355 (2008).
46. Lin, A. C., Bygrave, A. M., de Calignon, A., Lee, T. & Miesenböck, G. Sparse, decorrelated odor coding in the mushroom body enhances learned odor discrimination. *Nat. Neurosci.* **17**, 559–568 (2014).
47. Rao, R. P. N. & Ballard, D. H. Predictive coding in the visual cortex: a functional interpretation of some extra-classical receptive-field effects. *Nat. Neurosci.* **2**, 79–87 (1999).
48. Duchaine, B. & Yovel, G. A Revised Neural Framework for Face Processing. *Annu. Rev. Vis. Sci.* **1**, 393–416 (2015).
49. Kietzmann, T. C., Swisher, J. D., König, P. & Tong, F. Prevalence of Selectivity for Mirror-Symmetric Views of Faces in the Ventral and Dorsal Visual Pathways. *J. Neurosci.* **32**, 11763 (2012).
50. Anzellotti, S. & Caramazza, A. The neural mechanisms for the recognition of face identity in humans. *Front. Psychol.* **5**, (2014).
51. Ramírez, F. M. Orientation Encoding and Viewpoint Invariance in Face Recognition: Inferring Neural Properties from Large-Scale Signals. *The Neuroscientist* **24**, 582–608 (2018).
52. Tsantani, M. *et al.* FFA and OFA encode distinct types of face identity information. *J. Neurosci.* JN-RM-1449-20 (2021) doi:10.1523/JNEUROSCI.1449-20.2020.
53. Haxby, J. V., Hoffman, E. A. & Gobbini, M. I. The distributed human neural system for face perception. *Trends Cogn. Sci.* **4**, 223–233 (2000).
54. Rossion, B., Dricot, L., Goebel, R. & Busigny, T. Holistic Face Categorization in Higher Order Visual Areas of the Normal and Prosopagnosic Brain: Toward a Non-Hierarchical View of Face Perception. *Front. Hum. Neurosci.* **4**, (2011).
55. Fan, X., Wang, F., Shao, H., Zhang, P. & He, S. The bottom-up and top-down processing of faces in the human occipitotemporal cortex. *eLife* **9**, e48764 (2020).
56. Srinivasan, M. V., Laughlin, S. B. & Dubs, A. Predictive Coding: A Fresh View of Inhibition in the Retina. *Proc. R. Soc. Lond. B Biol. Sci.* **216**, 427–459 (1982).
57. Kumar, S., Kaposvari, P. & Vogels, R. Encoding of Predictable and Unpredictable Stimuli by Inferior Temporal Cortical Neurons. *J. Cogn. Neurosci.* **29**, 1445–1454 (2017).

58. Tang, M. F., Smout, C. A., Arabzadeh, E. & Mattingley, J. B. Prediction error and repetition suppression have distinct effects on neural representations of visual information. *eLife* **7**, e33123 (2018).
59. Bein, O., Gasser, C., Amer, T., Maril, A. & Davachi, L. Predictions transform memories: How expected versus unexpected events are integrated or separated in memory. *Neurosci. Biobehav. Rev.* **153**, 105368 (2023).
60. Chang, L., Egger, B., Vetter, T. & Tsao, D. Y. Explaining face representation in the primate brain using different computational models. *Curr. Biol.* **31**, 2785-2795.e4 (2021).
61. Holm, S. A Simple Sequentially Rejective Multiple Test Procedure. *Scand. J. Stat.* **6**, 65–70 (1979).
62. Bastos, A. M. *et al.* Canonical Microcircuits for Predictive Coding. *Neuron* **76**, 695–711 (2012).
63. van Kerkoerle, T. *et al.* Brain mechanisms of reversible symbolic reference: a potential singularity of the human brain. (2023) doi:10.1101/2023.03.04.531109.
64. Dubois, J., de Berker, A. O. & Tsao, D. Y. Single-Unit Recordings in the Macaque Face Patch System Reveal Limitations of fMRI MVPA. *J. Neurosci.* **35**, 2791–2802 (2015).

Reviewer #1 (Remarks to the Author):

The authors convincingly answered all my questions and I have no further comments.

Reviewer #2 (Remarks to the Author):

I thank the authors for the detailed and thoughtful response to my comments. The authors have addressed all of my concerns.

Reviewer #3 (Remarks to the Author):

The authors addressed most of my concerns in the previous round of review. I have one remaining question:

In the Methods it says: "baseline durations were jittered between 1.5, 3.5 and 5.5 s for intra-pair and 5.5, 7.5 and 9.5 s for inter-pair intervals". Therefore, there are longer baseline intervals before the "context-free" images than before the successor images, and as a result, stronger adaptations may occur for the successor images due to shorter intervals. I would not expect the neurons to adopt the representation of a higher order area due to adaptation, but the results in Figure 2 could be partially contributed by such a mechanism. Ideally, a control experiment in untrained monkeys could address this issue. At the very least, the authors should discuss whether and how adaptation might contribute to the results presented in the current manuscript.

FINAL REVIEWER COMMENTS and RESPONSES

Nature Communications, 27th May 2024

Tarana Nigam and Caspar M. Schwiedrzik

Reviewer #1 (Remarks to the Author):

The authors convincingly answered all my questions and I have no further comments.

We are happy to hear that we managed to answer all the questions from the reviewer and thank them for providing constructive feedback.

Reviewer #2 (Remarks to the Author):

I thank the authors for the detailed and thoughtful response to my comments. The authors have addressed all of my concerns.

We thank the reviewer for her/his constructive comments and for appreciating our efforts in answering them. We are happy to hear that we succeeded in resolving all outstanding questions.

Reviewer #3 (Remarks to the Author):

The authors addressed most of my concerns in the previous round of review. I have one remaining question:

In the Methods it says: “baseline durations were jittered between 1.5, 3.5 and 5.5 s for intra-pair and 5.5, 7.5 and 9.5 s for inter-pair intervals”. Therefore, there are longer baseline intervals before the “context-free” images than before the successor images, and as a result, stronger adaptations may occur for the successor images due to shorter intervals. I would not expect the neurons to adopt the representation of a higher order area due to adaptation, but the results in Figure 2 could be partially contributed by such a mechanism. Ideally, a control experiment in untrained monkeys could address this issue. At the very least, the authors should discuss whether and how adaptation might contribute to the results presented in the current manuscript.

We are glad to hear that we managed to address all the questions and thank the reviewer for her/his input.

We also thank the reviewer for this question. We would like to first explain the reasoning behind this specific rapid event related design and then discuss the point about adaptation.

The specific design choices were made to find an optimal compromise between the ability to extract single trial BOLD responses, the maximal number of trials, and the number of breaks that the animals needed. Baseline durations with jitter between 1.5, 3.5 and 5.5 s for intra-pair and 5.5, 7.5 and 9.5 s for inter-pair intervals were chosen to optimally be able to extract single trial BOLD responses, taking into consideration the specific timing and refractoriness of the macaque BOLD response^{1,2}. Longer inter stimulus intervals have advantages, but also limit the overall number of trials within a run. Increasing the overall number of trials has been found to compensate for shorter ISIs³. However, we could not arbitrarily increase run duration a) because this increase the negative impact of scanner drift that increases with time, and b) because we needed to consider how long the animals were able to stably and reliably fixate.

Furthermore, we wanted to make sure there is not a very long duration between the predictor and successor so that the predictions are effective once the successor appears. A previous fMRI study⁴ using a statistical learning paradigm used 1, 3, 5 s as the inter stimulus interval within a pair and we oriented ourselves on this paper. Therefore, considering all the constraints, we decided to decrease the intra-pair baseline duration, while having a longer inter-pair baseline duration. We had taken additional consideration in the experimental design to reduce adaptation of the BOLD signal by spatially jittering the images. Nevertheless, we acknowledge the reviewer's point that indeed stronger adaptation could occur for the successor images. We also fully agree with the reviewer's assessment that stronger adaptation could not possibly lead to a systematic transfer of tuning properties between higher and lower face areas. Unfortunately, we do not have access to any untrained monkeys at the moment or will in the foreseeable future; hence, we acknowledge the point that the reviewer made and comment on this in the discussion section of the manuscript. We hope that this addition to the manuscript addresses the reviewer's point convincingly and that the reasoning behind our design choices are evident.

References

1. Pelekanos, V. *et al.* Rapid event-related, BOLD fMRI, non-human primates (NHP): choose two out of three. *Sci. Rep.* **10**, 7485 (2020).
2. Leite, F. P. & Mandeville, J. B. Characterization of event-related designs using BOLD and IRON fMRI. *NeuroImage* **29**, 901–909 (2006).
3. Soon, C.-S., Venkatraman, V. & Chee, M. W. L. Stimulus repetition and hemodynamic response refractoriness in event-related fMRI. *Hum. Brain Mapp.* **20**, 1–12 (2003).
4. Schapiro, A. C., Kustner, L. V. & Turk-Browne, N. B. Shaping of Object Representations in the Human Medial Temporal Lobe Based on Temporal Regularities. *Curr. Biol.* **22**, 1622–1627 (2012).